# LoRe: Adaptive Interaction-Evaluation Routing with Per-Step Interaction Budgets for Iterative Graph Solvers

**Jintao Li**[1]   **Yong-Yi Wang**[1]   **Zheng-An Wang**[1]   **Heng Fan**[2 3 1 4]

## Abstract

Diffusion-based neural solvers for combinatorial optimization repeatedly re-evaluate dense edge/factor interactions, making inference expensive in wall-clock time and often memory-bound at scale. Inspired by the computational methodologies of many-body physics, we introduce LoRe, a training-free, inference-time drop-in wrapper that enforces per-step interaction-evaluation budgeting: at each iteration, it evaluates only a fixed fraction of interactions by dynamically routing computation to high-conflict or high-uncertainty interactions, instead of using a fixed sparsification (e.g., static kNN graphs or static masks). Under fully inclusive end-to-end wall-clock accounting, LoRe substantially improves scalability on the Maximum Independent Set (MIS) problem, extending feasible inference more than $3\times$ beyond the baseline's out-of-memory limit, delivering a $\sim8\times$ speedup and a $\sim12\times$ peak-memory reduction, with solution quality preserved in this regime. Demonstrating cross-task generality on the large-scale Traveling Salesperson Problem (TSP) and zero-shot robustness to topology shifts, LoRe achieves a $\sim15\times$ speedup at $n = 1000$ with a $44\times$ memory reduction and competitive tour quality.

## 1. Introduction

Many real-world combinatorial decision systems invoke a solver as an inner-loop primitive, re-optimizing repeatedly as requests arrive and constraints evolve. Examples include real-time dispatch and routing in dynamic vehicle routing, datacenter or cluster scheduling under changing job arrivals, and network resource allocation reacting to time-varying congestion (Pillac et al., 2013; Verma et al., 2015; Kelly et al., 1998). In these deployments, a feasible solution must be produced quickly and then improved under strict latency and memory budgets, so the binding constraint is often the per-iteration compute/memory envelope rather than just the total number of refinement steps, a classic "anytime" requirement in deployed decision systems (Zilberstein, 1996).

Iterative neural solvers on graphs, particularly diffusion-based and GNN-based models, have achieved strong performance on combinatorial optimization (CO) problems (Bengio et al., 2021; Cappart et al., 2023; Mazyavkina et al., 2021; Sun & Yang, 2023; Sanokowski et al., 2024; Zhao et al., 2025), including canonical tasks like Maximum Independent Set (MIS) and the Traveling Salesperson Problem (TSP). However, their scalability is bottlenecked by a recurring cost pattern: each refinement step requires a dense evaluation of the entire interaction topology (e.g., all edges or factor pairs) to resolve conflicts (Scarselli et al., 2009; Gilmer et al., 2017; Hamilton et al., 2017). With a fixed horizon $T$ and dense message passing, the computational cost scales as $O(T|\mathcal{A}|)$, and peak memory grows linearly with the interaction set size $|\mathcal{A}|$. On large graphs, this full-support sweep often breaches the hardware envelope, leading to Out-Of-Memory (OOM) failures or unacceptable latency.

This bottleneck presents a dilemma. Reducing the number of steps $T$ (e.g., Distillation) does not lower the per-step peak memory. On the other hand, static spatial sparsification (e.g., fixed kNN graphs) reduces the per-step footprint but fails to capture the state-dependent nature of combinatorial conflicts. In iterative refinement, the "hotspots", regions with high conflict or uncertainty, drift over time. A fixed sparse support inevitably misses newly critical interactions, causing error accumulation and trajectory drift.

We observe that this challenge conceptually mirrors the Many-Body Problem in condensed matter physics. In strongly correlated systems (e.g., the Hubbard Model), computing exact interactions across an infinite lattice is intractable. Instead, methods like Cluster Dynamical Mean-Field Theory (C-DMFT) decompose the system into a *Clus-*

[1]Beijing Key Laboratory of Fault-Tolerant Quantum Computing, Beijing Academy of Quantum Information Sciences, Beijing, China [2]Beijing National Laboratory for Condensed Matter Physics, Institute of Physics, CAS, Beijing, China [3]Beijing Key Laboratory of Advanced Quantum Technology, Beijing, China [4]Hefei National Laboratory, Hefei, China. Correspondence to: Zheng-An Wang <wangza@baqis.ac.cn>, Heng Fan <hfan@iphy.ac.cn>.

*Proceedings of the 43rd International Conference on Machine Learning*, Seoul, South Korea. PMLR 306, 2026. Copyright 2026 by the author(s).

*ter* (a local region where strong correlations are resolved exactly) and a *Bath* (a background field that approximates the global environment). Crucially, the coupling between the cluster and the bath ensures that local accuracy is maintained within a global context.

Inspired by this physical insight, we propose **LoRe** (**Lo**cal **Re**compute), a *training-free*, *inference-time* protocol that operationalizes this Cluster-Bath decomposition for graph solvers. LoRe enforces a hard per-step operator budget by dynamically routing computation:

- **The Cluster:** At each step, LoRe identifies a time-varying subset of high-conflict interactions ($M_t$) to evaluate exactly.
- **The Bath:** The influence of the omitted interactions is approximated via a lightweight global recall signal, preventing the Cluster from becoming disconnected from the global state.

Unlike static sparsification, LoRe tracks the "drifting hotspots" of the solver trajectory. It functions as a drop-in wrapper that transforms a standard dense solver into a budget-aware dynamical system without retraining. Figure 1 summarizes the LoRe protocol.

We evaluate LoRe as a training-free, inference-time drop-in wrapper using fully inclusive end-to-end wall-clock measurements under an explicit and auditable accounting protocol (described in Section 3). Unless stated otherwise, all comparisons are against DIFUSCO (Sun & Yang, 2023) using the same codebase and pretrained checkpoints; LoRe only changes inference-time routing under the per-step budget.

The main contributions of this work are summarized as follows:

- **Conceptual Formulation:** We formalize per-step operator budgeting for iterative graph solvers, establishing a rigorous framework to constrain computational and memory envelopes within each refinement step.
- **The LoRe Protocol:** We introduce LoRe, a training-free runtime wrapper that operationalizes a physics-inspired Cluster-Bath decomposition, inducing temporal operator sparsity via dynamic, state-dependent routing.
- **Auditable Accounting:** We establish a fully inclusive, end-to-end accounting protocol, providing a transparent benchmark for resource-constrained inference.
- **Empirical Validation:** We demonstrate that dynamic routing dominates strong static alternatives under matched budgets. LoRe extends feasible inference $3\times$ beyond baseline out-of-memory limits on MIS, and delivers up to a $15\times$ speedup with a $44\times$ memory reduction on TSP.

We conclude by outlining the paper organization: Section 2 reviews related work; Section 3 formalizes per-step budget-ing and the auditable accounting protocol, and then instantiates the budgeted interface with **LoRe**; Section 4 reports experiments; and Section 5 concludes.

## 2. Related Work

Neural solvers for CO increasingly treat inference as an iterative refinement process on graphs, spanning learned constructive policies and improvement heuristics as well as dynamical solvers that repeatedly re-evaluate interactions across steps (Bengio et al., 2021; Cappart et al., 2023; Mazyavkina et al., 2021; Vinyals et al., 2015; Bello et al., 2017; Kool et al., 2019; Nazari et al., 2018; Khalil et al., 2017). A key practical consequence of this paradigm is that latency and memory can be dominated by repeated interaction evaluation rather than a single forward pass, particularly as graph size grows. To make the main efficiency levers explicit, Table 1 summarizes representative lines by their horizon, support pattern, and whether a hard per-step budget is explicitly enforced.

Diffusion-style backbones exemplify this trade-off. In CO, diffusion-based refinement has been instantiated in solvers such as DIFUSCO and DiffUCO, and further supported by scalable discrete diffusion samplers (Sun & Yang, 2023; Sanokowski et al., 2024; 2025). These methods can exhibit strong anytime behavior, but their step-wise update often involves dense message passing over the interaction set, making step-level footprint a binding constraint in large-scale deployments.

LoRe is designed to operate at this step level without retraining: it keeps the backbone parameters and refinement horizon fixed, and instead allocates expensive interaction evaluation under a hard per-step budget through state-dependent routing. Closely related neural-CO lines combine iterative refinement with alternative paradigms—adaptive solution expansion (COExpander (Ma et al., 2025)), structured denoising with step-wise variable selection (StruDiCO (Wang et al., 2025)), and generation-as-search test-time scaling (GenSCO (Li et al., 2025))—which similarly re-evaluate interactions across steps; LoRe's step-level budgeting is paradigm-agnostic, and we demonstrate transfer to T2TCO and COExpander wrappers in Section 4.

A complementary efficiency thread focuses on reducing total sampling cost, including training–test alignment, step compression, and faster samplers or distillation for diffusion models (Li et al., 2023; 2024; Ho et al., 2020; Song et al., 2021b;a; Lu et al., 2022; Salimans & Ho, 2022). These techniques primarily decrease the number (or cost) of refinement steps, and are therefore orthogonal to mechanisms that enforce a strict compute/memory envelope inside each step. LoRe targets this within-step constraint and can, in principle, be composed with step-reduction or sampler-acceleration

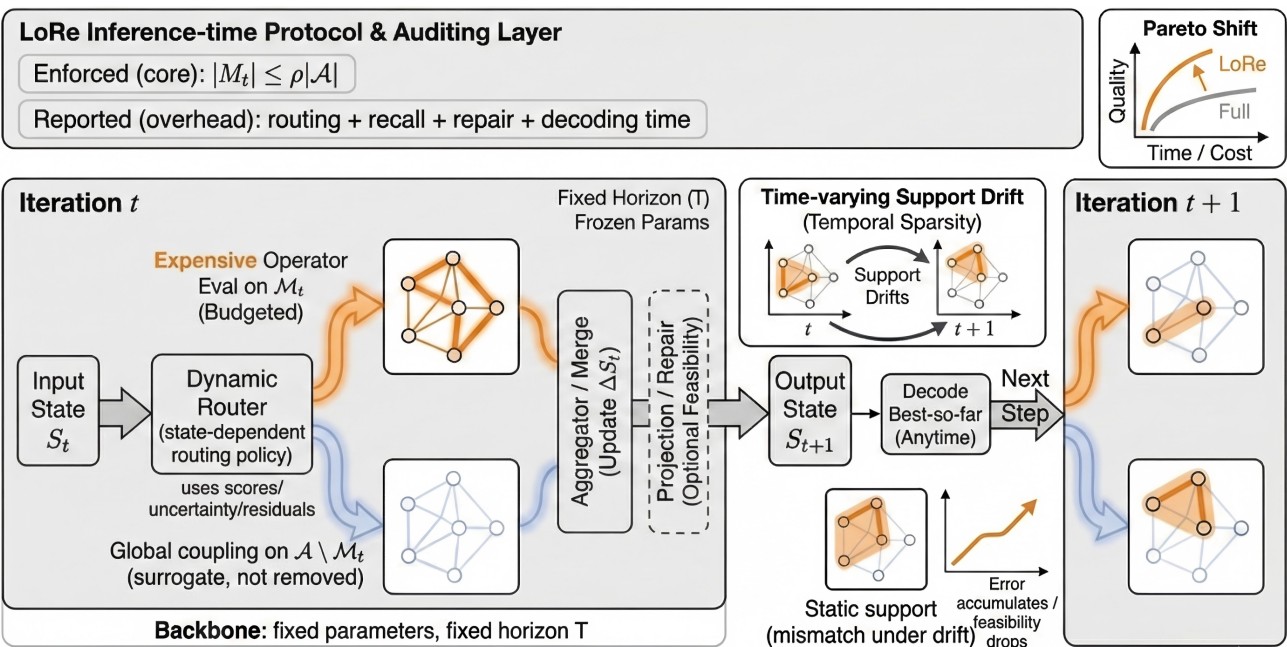

*Figure 1.* LoRe enforces *temporal operator sparsity* by routing per-step interaction evaluations to the interaction subset $M_t$ covered by time-varying high-conflict clusters under a hard budget $|M_t| \leq \rho|\mathcal{A}|$, optionally stabilized by lightweight recall and projection/repair.

methods.

Another line reduces per-step workload by restricting the interaction structure via static spatial sparsification, e.g., candidate graphs or fixed masks. Classic TSP pipelines use candidate edges and Lin–Kernighan style neighborhoods (Reinelt, 1991; Lin & Kernighan, 1973; Helsgaun, 2000), and analogous fixed supports also appear in learned solvers. Static supports are effective when high-impact interactions remain stable, but in iterative refinement the impact of interactions can shift along the trajectory; truncation error is therefore state-dependent and can accumulate if critical interactions are systematically omitted. This motivates allowing the expensive-evaluation support to vary over steps under an explicit budget.

A further related family of methods consists of outer-loop improvement paradigms, including destroy–repair and large neighborhood search (LNS), where a neighborhood is selected and re-optimized as a separate procedure (Shaw, 1998; Ropke & Pisinger, 2006; Pisinger & Ropke, 2010). Recent work also explores neural or diffusion-guided variants that leverage learned priors to propose neighborhoods or guide the improvement process (Hottung & Tierney, 2020; Feng et al., 2024).

LoRe adopts a different integration point: rather than wrapping refinement with an outer loop, it budgets expensive operator evaluations *inside every refinement step* of the solver dynamics, allocating interaction evaluations under a hard per-step envelope via state-dependent routing. Crucially,

the budgeted quantity here is per-step operator evaluation rather than neighborhood re-optimization; as a result, LoRe does not introduce an additional outer solver or change the backbone parameters or the refinement horizon.

There is a rich history of mapping CO problems to physical models, particularly Ising models and Hopfield networks, where energy minimization corresponds to optimization (Hopfield & Tank, 1985; Lucas, 2014). Graph Neural Networks themselves have been analyzed through the lens of mean-field inference in graphical models (Dai et al., 2016).

However, most prior works focus on the mapping formulation (e.g., constructing the Hamiltonian). Our work draws inspiration from the computational methodology of condensed matter physics—specifically Cluster Dynamical Mean-Field Theory (C-DMFT) (Maier et al., 2005). Instead of solving the full system or a purely local approximation, C-DMFT couples an exact local cluster with a mean-field bath. LoRe operationalizes this "Cluster-Bath" decomposition as a runtime routing protocol for neural solvers.

## 3. LoRe: Budgeted Runtime Routing for Iterative Graph Solvers

This section details the inference-time protocol. We first formalize the iterative solver dynamics and the per-step budget constraint. We then introduce a conceptual analogy to the Cluster-Bath decomposition in physical systems to motivate our operator design. Finally, we present the LoRe

*Table 1.* Solver-oriented positioning by efficiency lever. Representative references: DIFUSCO (Sun & Yang, 2023), COExpander (Ma et al., 2025), T2TCO (Li et al., 2023), Fast-T2T (Li et al., 2024).

| Work/line | Horizon | Support | Budget | Permanent |
|---|---|---|---|---|
| DIFUSCO | fixed | full | no | – |
| T2TCO / Fast-T2T | fewer | full | no | – |
| Static spatial sparsity | fixed | static | implicit | yes |
| Outer-loop refinement / LNS | varies | local | implicit | varies |
| COExpander | varies | varies | not explicit | varies |
| LoRe (ours) | fixed | time-varying | yes | no |

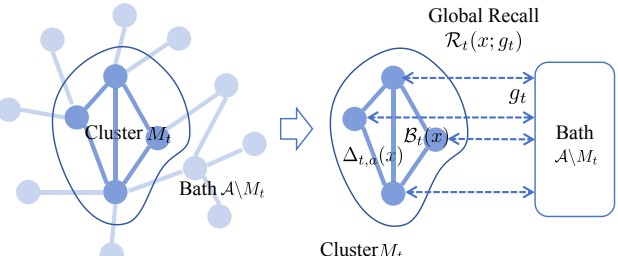

*Figure 2.* Inspired by C-DMFT, LoRe partitions the interaction graph at each step $t$ into two components. (Left) A high-conflict Cluster ($M_t$) selected via dynamic routing, and a low-conflict Bath ($\mathcal{A} \setminus M_t$). (Right) The influence of the omitted Bath is captured via a lightweight global signal $g_t$ and coupled to the Cluster through a global recall term $\mathcal{R}_t(x; g_t)$. $\mathcal{B}_t(x)$ denotes node-wise updates.

algorithm, which realizes this decomposition via dynamic routing.

## 3.1. Formulation: Iterative Refinement as Operator Evaluation

We view combinatorial optimization solvers as discrete dynamical systems. Let $x^t \in \mathbb{R}^{n \times d}$ denote the latent state of $n$ variables at step $t$. The refinement dynamics (e.g., diffusion denoising or GNN message passing) are governed by an operator $\mathcal{T}_t$ acting over the interaction graph $\mathcal{G} = (\mathcal{V}, \mathcal{A})$:

$$x^{t+1} = \Pi_t\big(\mathcal{T}_t(x^t; \mathcal{A})\big), \qquad t = 0, \dots, T-1. \quad (1)$$

where $\mathcal{T}_t$ aggregates information primarily through the dense interaction set $\mathcal{A}$ (e.g., edges in MIS or candidate moves in TSP), and $\Pi_t$ represents low-cost stabilization and feasibility handling (e.g., bounding logits, projection, decoding/repair). The computational bottleneck lies in $\mathcal{T}_t$, which can be written in a full interaction form:

$$\mathcal{T}_t(x; \mathcal{A}) = \mathcal{B}_t(x) + \sum_{a \in \mathcal{A}} \Delta_{t,a}(x), \quad (2)$$

where $\mathcal{B}_t$ captures node-wise constraints (independent evolution), while $\Delta_{t,a}$ represents conflict interactions (e.g., adjacent nodes in MIS). In dense graphs, evaluating interactions over the full set $\mathcal{A}$ scales as $O(|\mathcal{A}|)$, which becomes prohibitive for large instances. We enforce a per-step operator budget $\rho \in (0, 1]$, constraining the solver to evaluate only a subset $M_t \subseteq \mathcal{A}$ with $|M_t| \le \rho|\mathcal{A}|$.

## 3.2. Conceptual Intuition: The Cluster-Bath Decomposition

To approximate the dense operator in Eq. (2) under a strict per-step budget without discarding global consistency, we draw a high-level conceptual analogy to the computational methodologies developed for strongly correlated systems in condensed matter physics, specifically C-DMFT (Kotliar et al., 2001; Potthoff et al., 2003; Biroli et al., 2004; Maier et al., 2005).

In condensed matter physics, many-body interactions arise when particles strongly influence one another across a lattice, making it computationally intractable to treat them independently or evaluate exact global forces. Conceptually, CO exhibits a structurally identical bottleneck. The variables in CO are strongly coupled by task-specific constraints, such as adjacency exclusions in MIS or degree constraints in TSP. These constraints act precisely like physical many-body interactions ($\Delta_{t,a}$ in Eq. (2)): they dynamically form highly correlated "hotspots" where conflicts must be resolved exactly, while the rest of the graph settles into a stable background state.

C-DMFT sidesteps this bottleneck by partitioning the system into a *Cluster* (where intense short-range interactions are resolved exactly) and a surrounding *Bath* (which approximates the long-range global environment as a simplified background field).

We explicitly emphasize that LoRe does not establish a formal mathematical equivalence to quantum physical models, nor does it simulate real electron dynamics. Instead, we adapt C-DMFT's local-exact/global-approximate decomposition as an algorithmic blueprint. This physical intuition motivates our runtime protocol to dynamically isolate high-conflict interaction hotspots (the Cluster) from the stable background relations (the Bath), a structural pattern that LoRe operationalizes for graph solvers (Fig. 2).

## 3.3. LoRe: Budgeted Operator with Dynamic Routing

Inspired by the *Cluster-Bath* decomposition, LoRe partitions the interaction graph $\mathcal{A}$ at each step $t$ into a high-conflict subset $M_t$ and a low-conflict complement $\mathcal{A} \setminus M_t$. We replace the full operator in Eq. 2 with a budgeted operator:

$$\tilde{\mathcal{T}}_t(x; M_t, g_t) = \mathcal{B}_t(x) + \sum_{a \in M_t} \Delta_{t,a}(x) + \mathcal{R}_t(x; g_t), \quad (3)$$

where explicit, heavy computation is strictly restricted to the routed interaction subset $M_t$. This operator design directly instantiates the *Cluster-Bath* blueprint through a standard graph-neural lens:

- $\mathcal{B}_t(x)$ represents the independent node-wise evolution. It handles local state updates, conceptually mirroring isolated vertex dynamics rather than complex multi-node interactions.
- $\sum_{a \in M_t} \Delta_{t,a}(x)$ performs exact interaction evaluations exclusively over the dynamically selected high-conflict hotspots. This corresponds to the *Cluster*, where severe structural conflicts must be resolved precisely.
- $\mathcal{R}_t(x; g_t)$ introduces a lightweight global recall signal over the omitted interaction regions ($\mathcal{A} \setminus M_t$). Acting as a *mean-field coupling*, it approximates the *Bath* to connect the local *Cluster* to the global environment, ensuring the subgraph does not become structurally isolated.

Since the critical conflicts and computational hotspots drift continuously during the iterative refinement process, a static edge selection strategy would lead to catastrophic truncation error accumulation. LoRe mitigates this via a dynamic routing protocol, ensuring that $M_t$ adaptively tracks state-dependent importance across the solver trajectory. The overall operational protocol consists of three components:

**i. Dynamic Routing (Cluster Selection):** We employ a proxy score $s_{t,a} = S_t(a; x^t, x_{\text{prev}})$ to identify critical interactions. The active set $M_t$ combines a small static skeleton of structurally important interactions (fixed across $t$) with a dynamic hotspot set selected by the proxy score, under the strict constraint $|M_t| = B = \lfloor \rho|\mathcal{A}| \rfloor$. The scoring function prioritizes interactions with (a) high endpoint uncertainty (i.e., edges whose adjacency conflicts are not yet resolved) and (b) high temporal instability, ensuring that the dynamic allocation targets the largest residual contributors. To amortize routing overhead, $M_t$ is refreshed every $R$ steps.

**ii. Optional Global Recall:** To prevent the exact evaluation subgraphs from becoming disconnected from the global context, we compute a low-cost global background signal $g_t$ from the omitted interaction regions:

$$g_t = \text{Pool}_t(x^t; \mathcal{A} \setminus M_t), \quad \mathcal{R}_t(x^t; g_t) = U_t([x^t, g_t]). \quad (4)$$

We implement $U_t$ as a parameter-free coverage-weighted interpolation: $U_t([x^t, g_t])_i = \alpha_i x_i^t + (1 - \alpha_i)g_{t,i}$, where $\alpha_i = d_i(M_t)/d_i(\mathcal{A})$ represents the fraction of node $i$'s interactions evaluated exactly. This term (with $\mathcal{O}(N)$ complexity) acts as a contextual correction that stabilizes the optimization trajectory under tight budgets. The main experiments evaluate the pure-LoRe configuration (with recall disabled) to isolate the direct routing contribution; however, this parameter-free, optional stabilizer (requiring only one

**Algorithm 1** LoRe Inference (MIS)

---

**Require:** graph $G=(V,E)$, pretrained model $\mathcal{M}_\theta$, budget ratio $\rho$, skeleton ratio $\gamma$, refresh interval $R$, total steps $T$
1: **Precompute (run once):**
2: $\quad x^T \sim \mathcal{N}(0, I_n); \quad x_{\text{prev}} \leftarrow x^T$
3: $\quad B \leftarrow \lfloor \rho|E| \rfloor$
4: $\quad E_{\text{skel}} \leftarrow \text{Top}_{\lfloor \gamma B \rfloor} \{ \deg(i) + \deg(j) \; : \; (i,j) \in E \}$
5: **Sampling (per step $t$):**
6: **for** $t = T, T-1, \ldots, 1$ **do**
7: $\quad$ **if** $t = T$ **or** $t \bmod R = 0$ **then**
8: $\quad\quad E_{\text{hot}} \leftarrow \text{Top}_{B-|E_{\text{skel}}|} \{ S_t(e; x^t, x_{\text{prev}}) \; : \; e \in E \setminus E_{\text{skel}} \}$ $\qquad \triangleright S_t$: see §3.4
9: $\quad\quad M_t \leftarrow E_{\text{skel}} \cup E_{\text{hot}}$
10: $\quad$ **else**
11: $\quad\quad M_t \leftarrow M_{t+1}$
12: $\quad$ **end if**
13: $\quad \epsilon^t \leftarrow \mathcal{M}_\theta(x^t, t, M_t)$
14: $\quad x_{\text{prev}} \leftarrow x^t$
15: $\quad x^{t-1} \sim q(x^{t-1} \mid x^t, \epsilon^t)$
16: **end for**
17: **return** GreedyDecode$(x^0, G)$

---

cached tensor and no retraining) can be seamlessly enabled to further smooth trajectories in ultra-low budget regimes.

**iii. Fixed Projection/Repair and Greedy Decoding:** We keep the post-processing operator $\Pi_t$ (including decoding and feasibility repair) completely identical across all compared methods to ensure a rigorous and fair evaluation. In our target tasks, decoding and repair are designed to be standard and lightweight (consisting of a greedy pass followed by task-specific validity checks). Consequently, the recorded end-to-end acceleration stems entirely from alleviating the expensive interaction-evaluation bottleneck at the runtime level, rather than altering post-processing complexity.

Algorithm 1 summarizes the MIS instantiation; the TSP variant is described in Section 3.4 below.

**Error bound (informal).** Under a local Lipschitz assumption on the composed map $\Pi_t \circ \mathcal{T}_t$, the trajectory error $e_t = \|\tilde{x}^t - x^t\|$ between the budgeted and full updates satisfies

$$e_{t+1} \leq L_t \cdot e_t + \|\delta_t\|, \quad (5)$$

where the per-layer Lipschitz factor satisfies $L_t \leq \|A\|_2 \|W_t\|_2$, depending on graph structure through the adjacency spectral norm (bounded by $\Delta_{\max}$ for unweighted graphs), and the per-step residual decomposes via the triangle inequality as $\|\delta_t\| \leq \epsilon_t(\rho) + \|r_t\|$ with $\epsilon_t(\rho)$ the omitted-message mass and $\|r_t\|$ the recall approximation

error. LoRe's routing keeps $\epsilon_t(\rho)$ tight by sending high-impact interactions to the Cluster; empirically, the omitted (Bath) interactions remain near-certain throughout the sampling trajectory, so $\epsilon_t(\rho)$ stays negligible and $e_t$ does not accumulate appreciably (full derivation and measurements in Appendix A).

### 3.4. Task instantiations

We instantiate $\mathcal{A}$, score proxies $S_t$, and recall pooling for the tasks studied.

**MIS (primary).** Given $G = (V, E)$, set $\mathcal{A} = E$ and represent $x^t \in [0,1]^{|V|}$ as relaxed vertex indicators. We instantiate edge scores combining (i) endpoint uncertainty and (ii) temporal instability; one canonical form is

$$
\begin{aligned}
S_t\big((i,j); x^t, x_{\mathrm{prev}}\big) &= u_i\, u_j \\
&+ \lambda_{\mathrm{stab}} \left( |x_i^t - x_{\mathrm{prev},i}| + |x_j^t - x_{\mathrm{prev},j}| \right),
\end{aligned}
\tag{6}
$$

where the node uncertainty $u_i = 1 - |2x_i^t - 1|$ is maximal at $x_i^t = \frac{1}{2}$ (an undecided vertex), and $x_{\mathrm{prev}}$ is the state from the previous refinement step. This score is $O(|E|)$ to compute at refresh steps and highlights interactions that would otherwise cause truncation spikes. Framework-specific instantiations may use closely related node-level proxies based on the same two factors. Projection/repair uses the same greedy decode routine across all compared variants.

**TSP (secondary).** Let $\mathcal{A}$ be the interaction set used by the backbone (dense candidates or a pre-constructed candidate graph). LoRe routes expensive evaluation to a time-varying subset $M_t \subseteq \mathcal{A}$ without modifying $\mathcal{A}$ itself, highlighting the distinction between temporal operator sparsity (runtime routing) and static spatial sparsification (candidate graphs). Decoding/repair is kept identical across variants (greedy decode plus standard tour repair, with 2-opt where applicable).

## 4. Experiments

We evaluate **LoRe** as a training-free inference-time wrapper that enforces a hard per-step budget on interaction evaluations. Unless stated otherwise, all methods use the same DIFUSCO implementation and pretrained checkpoints, with identical refinement horizon and decoding/repair; LoRe modifies only the per-step active interaction set via runtime routing. All reported wall-clock time and peak GPU memory follow a fully inclusive end-to-end accounting protocol, covering diffusion steps and post-processing.

Experiments run on Ubuntu 24.04 with dual AMD EPYC 9654 CPUs and NVIDIA RTX PRO 6000 GPU (96GB). When randomness is present, we repeat runs with different seeds and report mean±std.

**Roadmap and central hypothesis.** Our central hypothesis is that enforcing a hard per-step interaction budget can improve end-to-end efficiency *at scale* while preserving stable inference behavior, provided that the evaluated support is re-routed as the state evolves. Table 2 summarizes the results across tasks. We build evidence for this claim in three stages:

- **Scalability and Feasibility (Secs. 4.1 and 4.2).** We quantify scaling on large MIS instances, explicitly probing the baseline out-of-memory (OOM) boundary, and verify cross-task generalization on TSP.
- **Mechanism and Necessity (Sec. 4.3).** We isolate routing under matched per-step budgets to test the necessity of state-dependent re-routing, contrasting dynamic versus static supports and greedy variants.
- **Deployment Behavior (Secs. 4.4 and 4.5).** We characterize sensitivity to LoRe's hyperparameters under a single untuned configuration and evaluate robustness to topology shift without retraining.

### 4.1. Scalability and Memory Feasibility on Large MIS Instances

We investigate whether enforcing a hard per-step interaction budget effectively improves end-to-end scaling and extends memory feasibility as graph size increases. Following the accounting protocol in Sec. 4, we sweep $n$ to explicitly probe the baseline out-of-memory (OOM) boundary, reporting both wall-clock time and peak GPU memory (Fig. 3(a)). Since aggregate metrics are provided in Table 2, we focus here on the scaling trends and feasibility limits.

Figure 3(a) illustrates divergent scaling trends in both memory and runtime. The full-support baseline rapidly saturates memory, encountering OOM at $n \approx 20$k nodes (its largest feasible scale is $n = 15$k), whereas LoRe remains feasible up to $n = 50$k (Table 2). Runtime performance exhibits a similar divergence: speedup increases with instance size, rising from $\sim 2\times$ (1k) to $\sim 8\times$ (10k–15k). This confirms that the routing overhead is effectively amortized as the dominant interaction-evaluation workload is reduced by the per-step budget. Notably, at 15k nodes (the largest feasible baseline scale), LoRe achieves an $8.16\times$ end-to-end speedup with 1.01 retention while reducing peak memory by $\sim 12\times$ (86.7 GB $\rightarrow$ 7.32 GB, see Table 2).

In summary, these results validate the scalability hypothesis: bounding expensive interaction evaluations transforms the scaling profile, improving practical feasibility and efficiency at scale rather than merely yielding a constant-factor acceleration.

*Table 2.* **Main Results: Scalability, Robustness, and Cross-Framework Generality.** Fully inclusive end-to-end wall-clock time and peak GPU memory. $N$: number of evaluation runs. Speedup = mean of per-instance time ratios (base/LoRe); Retention = ratio of per-instance solution-quality scores (LoRe/base for MIS set size; base/LoRe for TSP tour length; $\geq 1$ means LoRe matches or beats the baseline in either case); both reported as mean±std over the $N$ instances. Mem↓: base/LoRe peak-memory ratio. OOM: baseline exceeds the 96 GB GPU (relative metrics undefined).

| Fw. | Task | Setting | Time (s) LoRe/base | Mem (GB) LoRe/base | Mem↓ | N | Speedup (×) | Retention |
|---|---|---|---|---|---|---|---|---|
| **DIFUSCO — MIS: size scaling and OOM boundary (ER, $p$=0.05)** | | | | | | | | |
| DIFUSCO | MIS | $n$=1k | 7.9 / 17.3 | 0.07 / 0.42 | 5.7× | 5 | 2.19 ± 0.03 | 0.815 ± 0.048 |
| DIFUSCO | MIS | $n$=2k | 10.8 / 69.3 | 0.18 / 1.58 | 8.6× | 5 | 6.42 ± 0.04 | 0.804 ± 0.034 |
| DIFUSCO | MIS | $n$=3k | 18.6 / 149 | 0.35 / 3.51 | 10.0× | 5 | 8.03 ± 0.03 | 0.835 ± 0.017 |
| DIFUSCO | MIS | $n$=5k | 52 / 406 | 0.88 / 9.68 | 11.0× | 5 | 7.79 ± 0.07 | 1.029 ± 0.031 |
| DIFUSCO | MIS | $n$=8k | 124 / 1030 | 2.15 / 24.7 | 11.5× | 5 | 8.28 ± 0.12 | 1.019 ± 0.014 |
| DIFUSCO | MIS | $n$=10k | 196 / 1601 | 3.31 / 38.6 | 11.7× | 5 | 8.16 ± 0.14 | 0.991 ± 0.044 |
| DIFUSCO | MIS | $n$=15k | 442 / 3604 | 7.32 / 86.7 | 11.9× | 3 | 8.16 ± 0.04 | 1.010 ± 0.013 |
| DIFUSCO | MIS | $n$=20k | 767 / OOM | 12.9 / OOM | – | 3 | – | – |
| DIFUSCO | MIS | $n$=30k | 1782 / OOM | 28.8 / OOM | – | 3 | – | – |
| DIFUSCO | MIS | $n$=50k | 4949 / OOM | 79.5 / OOM | – | 3 | – | – |
| **DIFUSCO — TSP (dense-graph mode; dense-trained $n$=100 checkpoint): scaling to larger $n$ (merge + 2-opt; end-to-end)** | | | | | | | | |
| DIFUSCO | TSP | $n$=100 | 0.61 / 0.34 | 0.03 / 0.08 | 2.7× | 20 | 0.55 ± 0.05 | 0.936 ± 0.021 |
| DIFUSCO | TSP | $n$=500 | 0.72 / 3.61 | 0.05 / 1.23 | 24.6× | 20 | 5.10 ± 0.39 | 0.953 ± 0.014 |
| DIFUSCO | TSP | $n$=1k | 0.94 / 13.6 | 0.11 / 4.83 | 43.9× | 20 | 14.61 ± 1.07 | 0.960 ± 0.006 |
| DIFUSCO | TSP | $n$=2k | 2.14 / 56.7 | 0.35 / 19.5 | 55.6× | 15 | 26.66 ± 2.12 | 0.974 ± 0.008 |
| DIFUSCO | TSP | $n$=3k | 4.42 / 125 | 0.73 / 43.4 | 59.4× | 15 | 28.46 ± 1.95 | 0.983 ± 0.005 |
| DIFUSCO | TSP | $n$=4k | 7.92 / 228 | 1.28 / 77.0 | 60.1× | 5 | 28.84 ± 0.31 | 1.008 ± 0.004 |
| DIFUSCO | TSP | $n$=5k | 12.7 / OOM | 1.99 / OOM | – | 5 | – | – |
| DIFUSCO | TSP | $n$=10k | 50.6 / OOM | 7.85 / OOM | – | 5 | – | – |
| DIFUSCO | TSP | $n$=30k | 492 / OOM | 70.4 / OOM | – | 3 | – | – |
| **DIFUSCO — MIS: topology-OOD generalization ($n$=2k, 100 graph–budget evaluations/family)** | | | | | | | | |
| DIFUSCO | MIS | ER → ER | 1.08 / 8.58 | 0.39 / 4.04 | 12.4× | 100 | 8.67 ± 2.44 | 0.991 ± 0.053 |
| DIFUSCO | MIS | ER → BA | 1.05 / 8.07 | 0.37 / 3.78 | 12.4× | 100 | 8.25 ± 2.16 | 0.756 ± 0.051 |
| DIFUSCO | MIS | ER → WS | 1.12 / 8.57 | 0.39 / 4.04 | 12.4× | 100 | 8.31 ± 2.24 | 1.002 ± 0.048 |
| **Cross-framework: LoRe with framework-specific routing adaptations, no backbone retraining** | | | | | | | | |
| T2TCO | MIS | $n$=5k (size-OOD) | 13.0 / 60.1 | 2.49 / 29.2 | 11.7× | 30 | 4.62 ± 0.02 | 0.989 ± 0.083 |
| T2TCO | TSP | $n$=1k | 1.20 / 13.77 | 0.136 / 4.83 | 35.6× | 30 | 11.49 ± 0.68 | 0.994 ± 0.011 |
| COExpander | MIS | $V$=2.4k | 0.60 / 3.92 | 0.49 / 5.15 | 10.4× | 30 | 6.54 ± 0.56 | 0.872 ± 0.033 |
| COExpander | MVC | $V$=2.4k | 0.44 / 2.93 | 0.49 / 5.13 | 10.4× | 30 | 6.60 ± 0.72 | 1.001 ± 0.001 |

## 4.2. Cross-Task Scaling on TSP

We further validate that the proposed budgeting mechanism generalizes to a distinct task (TSP) without retraining. Following the same DIFUSCO codebase and identical merge + 2-opt post-processing, we evaluate TSP in DIFUSCO's dense-graph mode (the full-support interaction set, sparse_factor=−1) using the dense-trained $n$=100 checkpoint. We scale the instance size from the in-distribution point ($n$=100) up to $n$=4000, and additionally probe an OOM-extension regime (up to $n$=30k) where the dense baseline is infeasible. As shown in Table 2 and Figure 3(b), the efficiency gains become increasingly pronounced with graph size, directly aligning with our design goal for large-scale inference.

While LoRe introduces minor overhead on small instances ($n$=100) where the baseline runtime is already negligible, it delivers substantial speedups and near-saturating mem-

ory reductions as the computational burden increases. At $n$=1000, LoRe reduces end-to-end time from 13.6s to 0.94s and peak memory from 4.83 GB to 0.11 GB, keeping the tour length within a few percent of the baseline (retention 0.94–1.01 across scales, with the gap narrowing as $n$ grows). A runtime breakdown confirms the source of these gains: at $n$=1000 the baseline spends 13.3s (98% of its 13.6s total) on the diffusion interaction-evaluation stage, whereas LoRe compresses this stage to 0.57s. Because the merge + 2-opt post-processing is shared and identical, the speedup stems directly from alleviating the interaction bottleneck rather than from post-processing.

## 4.3. Necessity of State-Dependent Routing

**Controlled Routing Ablation.** We explicitly investigate the routing mechanism to determine whether *state-dependent re-routing* is essential under a hard per-step budget, or if a static sparse support suffices. To eliminate confounders

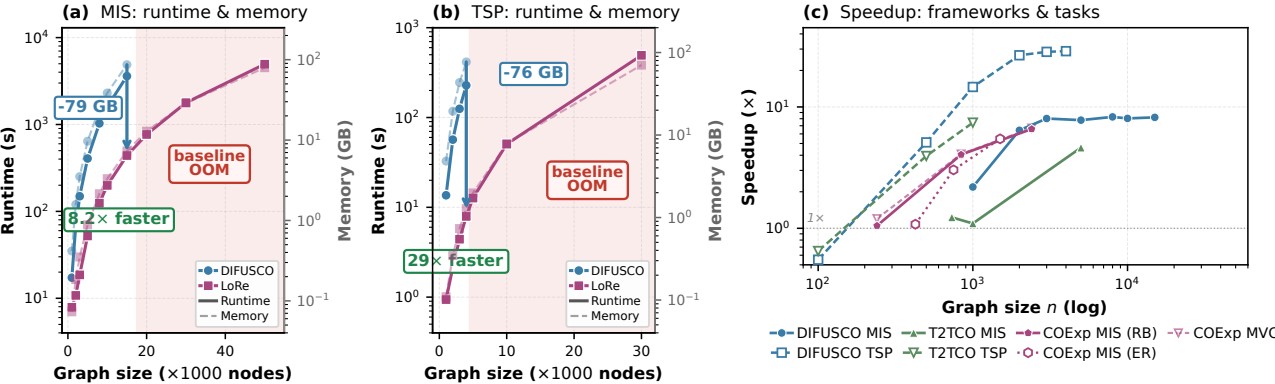

*Figure 3.* **Scaling and cross-framework performance.** (a) MIS: runtime and peak GPU memory vs. graph size. (b) TSP: same axes. (c) Speedup vs. graph size across multiple framework × task combinations. Red region marks baseline OOM.

arising from learning dynamics or checkpoint variance, we conduct a controlled ablation using a training-free iterative relaxation (conflict-descent style) on MIS instances, varying solely the edge selection rule.

We evaluate on 40 graphs: 20 Erdős–Rényi (ER, $p=0.05$) and 20 Barabási–Albert (BA, $m=3$) with $n \in \{500, 1000\}$. To ensure a rigorous comparison, all strategies share the same horizon ($T=100$), per-step budget ratio ($\rho=0.08$), and decoding/repair procedures. Furthermore, each instance uses an identical initialization $x^{(0)} \sim \mathrm{Uniform}(0.25, 0.75)$ shared across all variants.

*Table 3.* **Controlled Routing Strategies.** All variants enforce a strict per-step budget of $\lfloor \rho|E| \rfloor$; dynamic variants refresh the support every $R$ steps.

| Variant | Selection Rule | Refresh |
|---|---|---|
| LoRe | static skeleton + dynamic hotspots | every $R$ |
| Greedy-Conflict | top edges by conflict $x_i x_j$ | every $R$ |
| Greedy-Deg-Dyn | top edges by $(\deg_i + \deg_j)(x_i + x_j)$ | every $R$ |
| LoRe-Static | LoRe at $t=0$, then fixed | never |
| Greedy-Degree | top edges by degree sum $\deg_i + \deg_j$ | never |
| Random | uniform edge sampling | every $R$ |

Figure 4 demonstrates that dynamic re-routing consistently outperforms static supports under matched budgets. LoRe achieves rapid convergence in full-graph soft conflict energy $\mathcal{C}(x)$ and identifies repairable solutions early. In contrast, LoRe-Static exhibits early saturation, often failing to yield feasible solutions. Crucially, among dynamic strategies, LoRe is more stable than purely dynamic greedy edgewise selection. This underscores the advantage of *combining a static structural backbone with dynamic hotspots*, rather than relying on either component alone.

To further quantify why static masks fail, we track the

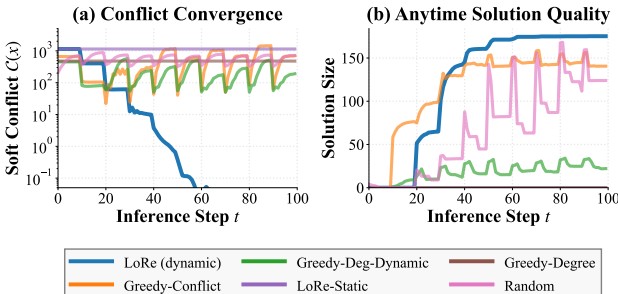

*Figure 4.* **Dynamic vs. Static Supports under Matched Budgets.** (a) Full-graph soft conflict energy $\mathcal{C}(x)$. (b) Anytime legal solution size after repair. Curves are averaged over 40 graphs; routing variants are detailed in Table 3.

temporal evolution of the selected edge set $M_t$. Under the default refresh interval $R = 10$, the overlap fraction $|M_t \cap M_{t+R}|/|M_t|$ between selected sets at consecutive refreshes is only 25.7% during the early diffusion stage (steps 1–10), reflecting a highly non-stationary conflict frontier as the global solution structure forms. This overlap increases to 56.4% in later refinement steps (steps 31–50), while remaining far from complete stability. This persistent temporal variation is consistent with the severe quality degradation of static support relative to LoRe's dynamic adaptation.

Collectively, these results confirm that the performance gains stem principally from the dynamic adaptation of the evaluation support as the state evolves.

## 4.4. Hyperparameter Sensitivity

We evaluate whether LoRe's default hyperparameters require per-setting tuning. Across all frameworks, tasks, graph densities, and problem scales considered in this paper, we use a single configuration ($\rho=0.08$, $R=10$, $\gamma=0.05$, $\lambda_{\mathrm{stab}}=0.5$) without task-specific tuning. To characterize the effect of each hyperparameter, we sweep them inde-

*Table 4.* **Hyperparameter Sensitivity.** A single default ($\rho$=0.08, $R$=10, $\gamma$=0.05, $\lambda_{\text{stab}}$=0.5) is used *without tuning* across all frameworks, tasks, and configs. We sweep the proxy-score weight $\lambda_{\text{stab}} \in [0, 5.0]$ across graph densities (DIFUSCO MIS, $n$=5000, 3 graphs/cell, $\rho$=0.08); each cell is mean quality retention (%).

| $\lambda_{\text{stab}}$ | $p$=0.05 | $p$=0.10 | $p$=0.15 |
|---|---|---|---|
| 0 | 99.7 | 105.1 | 96.8 |
| 0.25 | 99.1 | 104.0 | 97.6 |
| 0.5 | 99.1 | 105.1 | 97.6 |
| 1.0 | 98.8 | 103.4 | 97.6 |
| 2.0 | 98.8 | 105.7 | 98.4 |
| 5.0 | 99.7 | 102.9 | 96.8 |
| **Range** | 0.9 pp | 2.8 pp | 1.6 pp |
| **Speedup** | 6.3× | 6.6× | 6.6× |
| **Mem↓** | 11× | 12× | 12× |

pendently; Table 4 reports the sweep over $\lambda_{\text{stab}} \in [0, 5.0]$ across graph densities at $n$=5000.

Retention varies by at most 2.8 pp across $\lambda_{\text{stab}}$ at every density, while the speedup remains 6–7× and the peak-memory reduction remains 11–12×. Independent sweeps of the interaction budget $\rho$ and refresh interval $R$ show similar robustness, with retention varying by less than 3 pp over $\rho \in [0.05, 0.50]$ and less than 1.5 pp over $R \in [1, 50]$.

The few deviations are task- or backbone-dependent and consistent with the role of each hyperparameter. In particular, categorical-diffusion backbones can be more sensitive to infrequent refreshes.

Overall, LoRe is largely insensitive to its hyperparameters.

The default configuration transfers across settings without tuning, and when adjustment is needed the guideline is simple: a larger budget $\rho$ for constraint-dense tasks, and a shorter refresh interval $R$ for categorical-diffusion backbones.

### 4.5. Zero-Shot Robustness to Topology Shifts

We finally assess the **zero-shot generalization** of LoRe: can the efficiency gains persist under topology shifts without retraining the backbone? Using the ER-pretrained model, we evaluate Erdős–Rényi (ER), Barabási–Albert (BA), and Watts–Strogatz (WS) families over 100 graph–budget evaluations per family (Table 2, topology-OOD rows). LoRe delivers consistent ∼8× acceleration and ∼12× peak-memory reduction across all three families, with near-perfect retention on ER and WS (0.99 and 1.00); the only notable degradation is on the scale-free BA family (retention 0.76), where heavy-tailed degree distributions concentrate conflicts on hub nodes and make sparse budget allocation more challenging.

### Experimental Takeaways

Our experimental results validate LoRe as a scalable, training-free inference accelerator.

- **Scalability:** LoRe breaks the memory bottleneck of full-graph diffusion, extending feasibility to large instances where the baseline fails (OOM), while delivering increasing speedups.
- **Mechanism:** Controlled ablations confirm that dynamic, state-dependent re-routing is the key driver of performance, consistently outperforming static or greedy baselines.
- **Robustness:** The method generalizes zero-shot to new tasks (TSP) and topologies (OOD) and offers a controllable speed–quality trade-off via the budget parameter.

## 5. Conclusion

We formalize *per-step operator budgeting* for iterative neural solvers and cast it as a *runtime routing* problem. We present **LoRe**, a training-free, inference-time drop-in wrapper that enforces *temporal operator sparsity* via a time-varying active interaction set, leaving backbone checkpoints and decoding unchanged. Experiments show improved practicality at scale: LoRe reduces peak memory, shifts the MIS feasibility boundary beyond baseline OOM, and transfers to TSP with a scale-dependent crossover and bounded quality impact. Future work includes tighter stability/accuracy characterizations of budget-induced truncation and extending budgeted routing to broader solver families and tasks.

## Acknowledgments

This work was supported by National Natural Science Foundation of China ( U25A6009, 92265207, 12247168), MOST of China (2025YFE0217600), China Postdoctoral Science Foundation (2022TQ0036). We also thank the anonymous reviewers for their insightful comments, and Chang Liu for valuable discussions.

## Impact Statement

This paper presents work whose goal is to advance the field of machine learning. There are many potential societal consequences of our work, none of which we feel must be specifically highlighted here.

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

# A. Per-Step Error Bound: Full Derivation and Empirical Validation

This appendix gives the full derivation of the per-step error bound stated informally in the main text (Eq. 5), together with the trajectory measurements that validate its key assumption.

## A.1. Full derivation

The error recursion follows from the triangle inequality: at each step $t$, inserting the exact operator evaluated at the approximate state,

$$
\begin{aligned}
e_{t+1} &= \|\tilde{F}_t(\tilde{x}^t; \rho) - F_t(x^t)\| \\
&\leq \|F_t(\tilde{x}^t) - F_t(x^t)\| + \|\tilde{F}_t(\tilde{x}^t; \rho) - F_t(\tilde{x}^t)\|.
\end{aligned}
\tag{7}
$$

The first term captures propagated historical error; the second is the current-step truncation residual $\delta_t$.

**Dependence on graph structure ($L_t$).** Assuming a 1-Lipschitz activation such as ReLU, a single GNN aggregation layer (instantiating $\mathcal{T}_t$) computes $F_t(x) = \sigma(AxW_t)$. By the mean value inequality and matrix norm submultiplicativity,

$$
\begin{aligned}
\|F_t(\tilde{x}^t) - F_t(x^t)\| &= \|\sigma(A\tilde{x}^t W_t) - \sigma(Ax^t W_t)\| \\
&\leq \|A\|_2 \|W_t\|_2 \|\tilde{x}^t - x^t\| = L_t \cdot e_t.
\end{aligned}
\tag{8}
$$

Thus $L_t \leq \|W_t\|_2 \|A\|_2 \leq \|W_t\|_2 \Delta_{\max}$, where $\Delta_{\max}$ is the maximum node degree. This makes the dependence on graph structure explicit: denser regions with higher $\Delta_{\max}$ have a larger Lipschitz constant, leading to stronger error amplification per step.

**Dependence on edge budget $\rho$.** LoRe evaluates the Cluster $M_t$ (with $|M_t| = B = \lfloor \rho |\mathcal{A}| \rfloor$) exactly. The truncation residual decomposes via the triangle inequality as

$$
\|\delta_t\| \leq \sum_{a \in \mathcal{A} \setminus M_t} \|\mathrm{MSG}_{t,a}(\tilde{x}^t) - \mathrm{Pool}_t\| + \|r_t\| = \epsilon_t(\rho) + \|r_t\|,
\tag{9}
$$

where $\epsilon_t(\rho)$ is the omitted-message mass over $\mathcal{A} \setminus M_t$ and $\|r_t\|$ is the residual contributed by the optional recall mechanism of Eq. 4. As $\rho$ increases, $|\mathcal{A} \setminus M_t|$ shrinks, strictly decreasing $\epsilon_t(\rho)$. As verified empirically in Sec. A.2, the omitted interactions are highly deterministic, keeping $\|r_t\|$ tightly bounded.

**Grönwall bound.** Combining the above yields the recursion $e_{t+1} \leq L_t e_t + \|\delta_t\|$. Since both trajectories share the same initialization, $e_0 = 0$. Unrolling with $L = \max_t L_t$ and applying the geometric series,

$$
e_T \leq \sum_{k=0}^{T-1} L^{T-1-k} \|\delta_k\| \leq \frac{L^T - 1}{L - 1} \left( \epsilon_{\max}(\rho) + r_{\max} \right).
\tag{10}
$$

## A.2. Empirical validation

We measure the activity of omitted (Bath) versus retained (Cluster) interactions across the diffusion trajectory on TSP-100/500/1000 with $\rho=0.08$ and 50 DDIM steps, using three metrics at different levels of the output pipeline:

- **Entropy:** Shannon entropy of the edge probability.

- **Logit var:** variance of the pre-softmax log-odds, reflecting raw GNN output activity before softmax compression.

- **Prob var:** variance of the post-softmax probability, the final output used for decoding.

*Table 5.* Bath vs. Cluster interaction activity across the diffusion trajectory (DIFUSCO TSP, $\rho=0.08$, 50 DDIM steps). C = Cluster, B = Bath; each cell reports the range over all 50 steps.

| Scale | Entropy (C / B) | Logit var (C / B) | Prob var (C / B) |
|---|---|---|---|
| TSP-100 | $0.017$–$0.20$ / $2.9\times10^{-7}$ | $42.9$–$106$ / $4.9$–$5.7$ | $3.6\times10^{-2}$–$1.1\times10^{-1}$ / $1.1\times10^{-15}$ |
| TSP-500 | $0.038$–$0.13$ / $3.3\times10^{-7}$ | $26.4$–$31.9$ / $3.9$–$9.3$ | $1.9\times10^{-2}$–$4.3\times10^{-2}$ / $6.3\times10^{-15}$ |
| TSP-1000 | $0.034$–$0.037$ / $3.5\times10^{-7}$ | $16.3$–$16.9$ / $5.0$–$5.3$ | $6.0\times10^{-3}$–$7.6\times10^{-3}$ / $9.5\times10^{-16}$ |

Bath interactions maintain near-zero entropy throughout all $50$ steps across all scales—the model has already reached a near-certain decision about each Bath interaction. Pre-softmax logit variance confirms this is genuine, not a softmax artifact. Cluster interactions carry substantial entropy throughout, reflecting unresolved uncertainty that demands exact computation.

Together, the bound shows the total error depends on the omitted-message mass $\epsilon_{max}(\rho)$ and the recall error $r_{max}$, amplified by the structural factor $L$; the measurements confirm the omitted interactions are near-certain throughout the trajectory, which keeps both $\epsilon_t(\rho)$ and $\|r_t\|$ small.

## B. Absolute Quality Benchmarks and Classical Solvers

In the main text, we report quality retention relative to the dense backbone to strictly isolate the effect of budgeted inference. To properly contextualize the absolute performance of the neural solver, we provide supplementary benchmarks against classical strong heuristics (KaMIS for MIS and Concorde for TSP).

*Table 6.* Absolute performance benchmarks on large-scale MIS instances. The baseline (BL) represents the dense DIFUSCO solver. LoRe successfully extends feasible inference to $n = 20k$ where the baseline triggers Out-Of-Memory (OOM) on a 96 GB GPU.

| Graph Size ($n$) | KaMIS | BL | LoRe | Speedup | Mem (BL / LoRe) |
|---|---|---|---|---|---|
| $5k$ | 157 | 108 | 111 | $7.79\times$ | 9.7 GB / 0.88 GB |
| $10k$ | 169 | 126 | 119 | $8.05\times$ | 38.6 GB / 3.31 GB |
| $20k$ | – | OOM | 140 | – | OOM / 12.9 GB |

As demonstrated in Table 6, the neural baseline exhibits an out-of-distribution (OOD) quality gap relative to KaMIS in this large-scale regime. These absolute-quality results are provided for context: LoRe targets the scalability bottleneck of neural inference rather than replacing specialized classical solvers.

Crucially, at massive scales ($n \geq 20k$), the dense baseline entirely exhausts the 96 GB memory limit. LoRe, however, remains fully operational within 12.9 GB, delivering a valid independent set of size $140$. Furthermore, on the TSP-500 task, under a quality-prioritized configuration, LoRe attains a $5.76\%$ optimality gap relative to Concorde, within $\sim$1.8pp of the sparse baseline's $3.97\%$.

