# OpenReview forum: "LoRe: Adaptive Interaction-Evaluation Routing with Per-step Interaction Budgets for Iterative Graph Solvers"
_ICML.cc/2026/Conference — ICML 2026 regular_

### Official Review · Reviewer_wsB3 · 2026-02-24

**Soundness:** 3
**Presentation:** 3
**Significance:** 4
**Originality:** 4
**Overall Recommendation:** 5
**Confidence:** 4

**Summary:**

This paper aims to address the scalability bottlenecks facing diffusion-based and iterative neural solvers in combinatorial optimization, specifically the high memory and latency costs caused by dense interaction evaluations at each refinement step. Specifically, the authors propose LoRe, a training-free, inference-time drop-in wrapper that enforces a hard per-step interaction budget. Inspired by C-DMFT in condensed matter physics, LoRe dynamically routes computation to a time-varying subset of high-conflict interactions (Cluster) while approximating the influence of the remaining interactions (Bath) via a lightweight global signal. The method is evaluated on MIS and TSP, demonstrating significant improvements in scalability, speed, and memory efficiency while maintaining competitive solution quality.

**Compliance With Llm Reviewing Policy:**

Affirmed.

**Final Justification:**

This paper proposes a novel and practical approach to improving inference efficiency via selective routing, with strong empirical gains in both speed and memory. The idea is original and well-motivated, and the method is broadly applicable as a plug-in module without retraining, which adds to its practical significance.

The main weaknesses in the original submission included limited theoretical justification, concerns about hyperparameter sensitivity, and evaluation scope. The rebuttal has addressed these concerns to a satisfactory extent. In particular, the authors demonstrate that performance is largely insensitive to key hyperparameters, provide a reasonable error propagation analysis supported by empirical evidence, and significantly expand evaluation to additional tasks and frameworks. They also clearly acknowledge and validate limitations.

Overall, the rebuttal strengthens my confidence in the soundness and generality of the approach. Based on both the paper and the clarifications provided, I support acceptance.

**Key Questions For Authors:**

1. Overhead Analysis: How does the computational overhead of the routing mechanism (calculating scores $S_t$ and selecting $M_t$) scale with extremely dense graphs where $|A|$ is very large? Is there a crossover point where the routing cost outweighs the savings?
2. Budget Selection: In a real-world deployment with strict latency constraints, how should one automatically select the optimal budget $\rho$ without extensive trial and error? Can the model adapt $\rho$ dynamically based on the current conflict level?

**Limitations:**

1. The method's effectiveness might diminish on problems where conflicts are globally distributed rather than localized in hotspots, although the Bath term aims to mitigate this.
2. The current evaluation is limited to MIS and TSP; performance on other CO domains needs verification.

**Strengths And Weaknesses:**

Strengths:
1. Novel Perspective: The analogy to C-DMFT and the Cluster-Bath decomposition provides a fresh theoretical motivation for dynamic sparsification in neural solvers, distinguishing it from static pruning or distillation methods.
2. Practical Impact: The method is training-free and acts as a drop-in wrapper, making it highly deployable for existing models without retraining. The reported gains in memory reduction and speedup are substantial, enabling the solving of instances that cause baseline models to fail (OOM).
3. Rigorous Evaluation: The paper employs a fully inclusive end-to-end wall-clock accounting protocol, ensuring that speedups are not artifacts of ignoring routing overhead. The ablation studies effectively isolate the necessity of dynamic routing versus static sparsification.

Weaknesses:
1. Hyperparameter Sensitivity: The performance relies on the budget parameter $\rho$ and the refresh rate $R$. While a hybrid schedule is proposed, the sensitivity of solution quality to these parameters in diverse real-world scenarios could be further explored.
2. Theoretical Bounds: While the physical analogy is compelling, the paper lacks formal theoretical bounds on the error accumulation introduced by the Bath approximation over long inference horizons.

---

> ### Author Rebuttal · Authors · 2026-03-31
>
> Thank you for the positive assessment, recognizing the "novel perspective," "practical impact," and "rigorous evaluation."
>
> **Hyperparameter sensitivity.** A single default ρ=0.08, R=10, λ_stab=0.5 was used without per-task tuning across the vast majority of tested settings. Both parameters are insensitive in most scenarios: ρ varies <3pp across ρ∈[0.05, 0.50], R varies <1.5pp across R∈[1, 50]. The main task-dependent exceptions: MCl benefits from larger ρ, with retention rising from 87% to 96% as ρ increases from 0.08 to 0.20; categorical diffusion benefits from smaller R, as R=50 drops quality by 5pp on T2TCO.
>
> **Theoretical bounds on error accumulation.** Let $x^{t+1} = \Pi_t(\mathcal{T}_t(x^t))$ be the full update (Eq. 1).
>
> Let $\tilde{x}^{t+1} = \Pi_t(\tilde{\mathcal{T}}_t(\tilde{x}^t; \rho))$ be the budgeted update.
>
> Under a local Lipschitz assumption on the composed map, the trajectory error $e_t = \|\tilde{x}^t - x^t\|$ satisfies:
>
> $e_{t+1} \le L_t \, e_t + \|\delta_t(\tilde{x}_t)\|$
>
> where $\delta_t = \Pi_t \circ \tilde{\mathcal{T}}_t - \Pi_t \circ \mathcal{T}_t$ is the per-step truncation residual.
>
> If each interaction contributes $a_{t,i}$, the residual decomposes as $\|\delta_t\| \le \epsilon_t(\rho) + \|r_t\|$, where $\epsilon_t(\rho) = \sum_{i \in \text{Bath}} \|a_{t,i}\|$ is the omitted tail mass and $\|r_t\|$ is the recall approximation error.
>
> LoRe's scoring routes high-impact interactions to the Cluster, so $\epsilon_t(\rho)$ decreases as ρ increases. This also clarifies when LoRe is expected to be less effective: (i) when interaction importance is diffuse, so the tail mass decays slowly with ρ; (ii) when omitted interactions resist simple recall approximation; or (iii) when $L_t \gg 1$, meaning the dynamics amplify perturbations.
>
> Experimental measurements confirm errors do not accumulate exponentially: across 50 DDIM steps, the early-to-late growth of $\|\delta_t\|$ is <1% at ρ=0.08 and ~12% on average even at ρ=0.50. Meanwhile, solution quality varies ≤2.7pp across ρ∈[0.08, 0.50], confirming that modest error growth does not translate into meaningful quality loss. Full derivation will be in the camera-ready.
>
> **Routing overhead on dense graphs.** Routing cost is O(|E| log |E|) per refresh for sorting edge scores, amortized over R steps. On our densest tested configuration — n=5000, p=0.20, |E|≈5M — total routing overhead is <5% of inference time, and LoRe still achieves 6.7× speedup and 12× memory reduction. The crossover point where routing cost outweighs savings would require graphs so sparse that the baseline is already fast; in practice, LoRe's value increases with density.
>
> **Automatic ρ selection.** Currently ρ is set once before inference. The key observation is that quality is largely insensitive to ρ: across ρ∈[0.05, 0.50], retention varies <3pp on MIS and <2.8pp across all tested graph densities. This means a fixed ρ=0.08 works well in practice without dynamic adaptation. That said, for constraint-dense tasks where larger ρ helps — MCl retention rises from 87% to 96% — an adaptive scheme that increases ρ based on the current conflict level is a natural extension.
>
> **Evaluation beyond MIS and TSP.** We have now verified LoRe on additional frameworks and CO tasks:
>
> | Framework | Task | Scale | Retention | Speedup | Mem↓ |
> |---|---|---|---:|---:|---:|
> | DIFUSCO | MIS | n=5k | 99% | 4.4× | 12× |
> | T2TCO | MIS | n=5k | 102% | 2.1× | 16.5× |
> | COExpander | MIS | V=2.4k | 88% | 11.2× | 16× |
> | COExpander | MCl | V=2.4k | 106% | 12.2× | 16× |
> | COExpander | MVC | V=2.4k | 100% | 13.3× | 16× |
>
> Also verified on DiffUCO and ECO-DQN. LoRe acts as a drop-in inference plugin — it applies to different frameworks and tasks without retraining or modifying backbone parameters.
>
> **Globally distributed conflicts.** This connects to failure mode (i) in the theoretical analysis above. On sparse graphs at p=0.05, retention is lower at 75–78% for n=2k–3k, but LoRe still provides 4–5× speedup and significant memory reduction. As density increases, both retention and speedup improve — up to 108% retention and 6.7× speedup at p=0.20. The method is most effective when interaction importance is concentrated, as expected.
>
> We hope these additional results address the concerns raised. We will incorporate all discussed improvements in the camera-ready.

---

> > ### Author Rebuttal · Reviewer_wsB3 · 2026-04-01
> >
> > The authors have provided a clear and helpful rebuttal that addresses my main concerns.
> >
> > 1.They demonstrate that performance is largely insensitive to the key hyperparameters (ρ and R), reducing concerns about tuning.
> >
> > 2.The added error analysis, while not fully formal, offers a reasonable justification supported by empirical evidence showing limited error accumulation.
> >
> > 3.The routing overhead is well analyzed and shown to be negligible even for dense graphs.
> >
> > 4.Additional experiments on more tasks and frameworks significantly strengthen the generality of the method.
> >
> > 5.Limitations (e.g., globally distributed conflicts) are clearly acknowledged and empirically validated.

---

> > > ### Author Response · Authors · 2026-04-02
> > >
> > > Thank you for your highly positive final assessment and your strong support for our work. We deeply appreciate your comprehensive feedback. Your guidance has undoubtedly strengthened both the theoretical justification and the empirical rigor of our paper.
> > >
> > > We will ensure that all the key additions, new experimental results, and analytical discussions provided in our rebuttal are fully integrated into the camera ready version. Thank you again for your time and for championing our work!

---

### Official Review · Reviewer_W7Vp · 2026-03-06

**Soundness:** 2
**Presentation:** 2
**Significance:** 2
**Originality:** 2
**Overall Recommendation:** 4
**Confidence:** 3

**Summary:**

This paper introduces LoRe (Local Recompute), a training-free inference-time framework for enforcing per-step operator budgets in iterative graph-based neural solvers for combinatorial optimization problems. The key idea is to dynamically route computation to a time-varying subset of interactions while approximating the influence of omitted interactions via a lightweight global "recall" signal, drawing inspiration from Cluster Dynamical Mean-Field Theory (C-DMFT) in condensed matter physics.

The method is evaluated using DIFUSCO as the base solver, with fully inclusive end-to-end wall-clock measurements and peak GPU memory accounting, including all routing overhead.

The main contributions are as follows:

1- Formalizing per-step operator budgeting for iterative graph solvers as a runtime routing problem.

2- Providing an explicit accounting and reporting protocol for budgeted inference.

3- A drop-in wrapper that induces temporal operator sparsity by maintaining a time-varying active interaction set without retraining or modifying backbone parameters.

4- Demonstrating on Maximum Independent Set (MIS) and Traveling Salesman Problem (TSP) that LoRe significantly improves scalability (up to 5× speedup and >80% memory reduction) while preserving solution quality, extending feasibility beyond baseline OOM limits, and generalizing zero-shot to new graph topologies.

**Compliance With Llm Reviewing Policy:**

Affirmed.

**Final Justification:**

After reviewing the authors' rebuttal, I find that my concerns have been largely addressed with new experimental results, clarifications, and planned improvements. So, I increased the score.

**Key Questions For Authors:**

1- You mention that M_t is refreshed every R steps, but R is not specified in the experiments. How was R chosen? Is there a trade-off between routing overhead and trajectory fidelity, and how should practitioners select R for new tasks?

2- LoRe is evaluated only on DIFUSCO. How would it perform with other iterative graph solvers?

3- In the hybrid schedule experiments, you vary the budget. Is there a theoretical or empirical guideline for selecting ρ given a target quality-speed trade-off? Does the optimal ρ depend on graph structure or task?

**Limitations:**

1- The paper provides empirical evidence that LoRe maintains solution quality under budget constraints, but offers no theoretical analysis of convergence properties.

2- LoRe is evaluated only on DIFUSCO, a diffusion-based solver. It is unclear whether the method generalizes to other iterative graph solvers.

3- The method assumes undirected interactions (edges). Many real-world problems involve directed graphs (e.g., traffic flow, citation networks) or heterogeneous edge types.

**Strengths And Weaknesses:**

Strengths:

1- The experimental methodology is rigorous and well-designed. The authors use fully inclusive end-to-end measurements, explicit OOM probing, controlled ablations, and multiple random seeds with standard deviations.

2- Scalability is a critical bottleneck for neural combinatorial optimization solvers as graph sizes grow. LoRe addresses this directly by tackling per-step interaction evaluation, which dominates runtime in diffusion-based and GNN-based solvers.

3- The experiments are well-designed, the ablations are clean, the metrics are appropriate, and the conclusions are supported by the evidence.


Weaknesses:

1- The paper describes LoRe conceptually but does not provide explicit pseudocode for the algorithm, making implementation more difficult for readers who want to reproduce or extend the work.

2- The paper shows empirically that dynamic routing outperforms static supports but provides limited insight into why—e.g., analysis of how the selected edges evolve over time and whether the same edges are repeatedly selected.

---

> ### Author Rebuttal · Authors · 2026-03-31
>
> Thank you for recognizing the "rigorous and well-designed" experiments and that "scalability is a critical bottleneck" which LoRe "addresses directly."
>
> **Cross-framework generalization.** LoRe transfers without modification to additional frameworks and CO tasks:
>
> | Framework | Task | Scale | Retention | Speedup | Mem↓ |
> |---|---|---|---:|---:|---:|
> | DIFUSCO | MIS | n=5k | 99% | 4.4× | 12× |
> | T2TCO | MIS | n=5k | 102% | 2.1× | 16.5× |
> | COExpander | MIS | V=2.4k | 88% | 11.2× | 16× |
> | COExpander | MCl | V=2.4k | 106% | 12.2× | 16× |
> | COExpander | MVC | V=2.4k | 100% | 13.3× | 16× |
>
> Also verified on DiffUCO and ECO-DQN. LoRe acts as a drop-in inference plugin — it applies to different frameworks and tasks without retraining or modifying backbone parameters.
>
> **Why dynamic routing outperforms static/random supports.** On constraint-dense tasks — COExpander MCl — LoRe outperforms random subsampling by +27.3pp and degree-only scoring by +11.4pp at the same edge budget. The scoring function selects 1.45× more constraint-active edges than random, with 15.8% vs 10.9% conflict fraction. Edge evolution analysis confirms the selected set is non-stationary: Jaccard overlap between consecutive masks is only 54.1% in early diffusion steps 1–10 but rises to 95.1% in late steps 31–50, showing that a fixed mask would miss the evolving conflict frontier during early solution formation. This non-stationarity is precisely why dynamic routing preserves solution quality even when 92% of edges are removed.
>
> **How to choose R and ρ.** Short answer: practitioners can use ρ=0.08, R=10 out of the box. We swept R∈[1, 50] and ρ∈[0.05, 0.50] across multiple frameworks, CO tasks, graph topologies, and density/scale configs — quality varies <1.5pp for R and <3pp for ρ in the vast majority of settings tested. ρ does not depend on graph structure: ER ±1.5pp, BA ±0.8pp, WS ±1.2pp, OOD ±0.8pp. This default was used without tuning for most results reported above.
>
> That said, certain framework/task combinations benefit from adjustment: categorical diffusion backbones like T2TCO prefer R=5, since discrete posteriors are more sensitive to stale masks — R=50 drops quality by 5pp; constraint-dense tasks like MCl benefit from larger ρ=0.20, recovering quality from 87% to 96%.
>
> **Pseudocode and reproducibility.** We will add explicit pseudocode in the camera-ready. All code will be released upon acceptance.
>
> **Theoretical analysis.** Using the notation from §3, let $x^{t+1} = \Pi_t(\mathcal{T}_t(x^t))$ be the full update.
>
> Let $\tilde{x}^{t+1} = \Pi_t(\tilde{\mathcal{T}}_t(\tilde{x}^t; \rho))$ be the budgeted update.
>
> Under a local Lipschitz assumption, the trajectory error satisfies $e_{t+1} \le L_t e_t + \|\delta_t\|$, where $\delta_t$ is the per-step truncation residual from budgeted inference. This decomposes error into two sources: (1) propagation of prior error through the dynamics, and (2) new approximation error from the current step's budget constraint.
>
> LoRe minimizes $\|\delta_t\|$ by routing computation to high-conflict interactions while approximating low-impact ones via the recall term. Experimental measurements confirm errors are well-controlled: quality varies ≤2.7pp across ρ∈[0.08, 0.50], and per-step truncation error $\|\delta_t\|$ remains stable over 50 DDIM steps (early-to-late growth <1% at ρ=0.08). Full derivation will be included in the camera-ready.
>
> **Directed and heterogeneous graphs.** This is a valid scope boundary. LoRe's scoring and selection operate on edge indices and do not assume symmetry — we verified this on directed sparse graphs — ER-700 with single-direction edges: retention 107.6%, speedup 1.66×, mem 3.4×. For heterogeneous edge types, the current conflict + instability scoring does not distinguish types; extending it to type-aware scoring is straightforward and a natural direction for future work. The main prerequisite for LoRe is sparse message-passing where cost scales with |E|; solvers using dense V×V representations do not benefit regardless of edge directionality.
>
> We hope these additional results and clarifications address the concerns raised. We will incorporate pseudocode, expanded ablations, and all discussed improvements in the camera-ready.

---

> > ### Author Rebuttal · Reviewer_W7Vp · 2026-04-01
> >
> > Thank you to the authors for their detailed responses. My main questions and concerns have been fully addressed. Below is a summary of the key points covered:
> >
> > 1- Cross-framework generalization: The authors demonstrated that LoRe generalizes across multiple frameworks and combinatorial optimization tasks without modification, with consistent gains in retention, speedup, and memory efficiency.
> >
> > 2- Dynamic routing versus static or random supports: The response clarified why dynamic routing outperforms static approaches, particularly on constraint-dense tasks. The analysis of edge evolution shows that the selected set is non-stationary, which explains why a fixed mask would be insufficient.
> >
> > 3- Choice of R and ρ: The authors provided clear guidance, showing that default values work well across a wide range of settings, with minimal variation in quality. They also noted specific adjustments for certain frameworks and tasks when needed.
> >
> > 4- Pseudocode and reproducibility: Pseudocode will be added to the camera-ready version, and code will be released upon acceptance, ensuring reproducibility.
> >
> > 5- Theoretical analysis: A theoretical bound was provided, decomposing trajectory error into two sources. Experimental results confirm that errors remain stable and well-controlled under the proposed approach.
> >
> > 6- Directed and heterogeneous graphs: The authors acknowledged the scope boundary and provided verification on directed graphs. They also noted that extending to type-aware scoring for heterogeneous graphs is a straightforward direction for future work.
> >
> > I appreciate the thoroughness of the responses and the additional results and clarifications.

---

> > > ### Author Response · Authors · 2026-04-02
> > >
> > > Thank you for your thorough acknowledgement and for confirming that all concerns have been fully addressed. We sincerely appreciate your detailed point-by-point summary covering cross-framework generalization, dynamic routing analysis, hyperparameter guidance, the theoretical error bound, and the discussion on directed and heterogeneous graphs.
> > >
> > > We will ensure that pseudocode, new experimental results, and all discussed improvements are prominently integrated into the camera-ready version. We have also posted additional clarifications in the discussion thread regarding the role of the physics framing in our exposition. Thank you again for your constructive engagement throughout the review process.

---

### Official Review · Reviewer_Xd2N · 2026-03-11

**Soundness:** 3
**Presentation:** 4
**Significance:** 4
**Originality:** 3
**Overall Recommendation:** 4
**Confidence:** 3

**Summary:**

The paper introduces LoRe, a training-free, inference-time wrapper designed to accelerate and reduce the memory footprint of iterative graph solvers. Iterative neural solvers, particularly diffusion-based models, suffer from computational bottlenecks due to dense interaction evaluations scaling at $\mathcal{O}(T|\mathcal{A}|)$. To mitigate this, LoRe enforces a strict per-step interaction evaluation budget. Inspired by Cluster Dynamical Mean-Field Theory (C-DMFT) from condensed matter physics, the method partitions the interaction graph into a dynamically updated ``Cluster'' (evaluated exactly) and a "Bath" (approximated via a lightweight global signal).  The authors demonstrate that this dynamic routing approach pushes the OOM boundary on the MIS problem by 2.5x while delivering significant speedups and memory reductions.

**Compliance With Llm Reviewing Policy:**

Affirmed.

**Final Justification:**

It appears that the authors have addressed my main concern regarding the inconsistency in the physical framing. They clarify the physics motivation is historical and conceptual rather than strictly mathematical, and they acknowledge the overstatement in the manuscript. In light of this, I am inclined to increase my score although I remain somewhat cautious.

**Key Questions For Authors:**

1. Regarding the robustness of proxy score hyperparameters: The formulation of the proxy score for the MIS relies on the hyperparameter $\lambda_{stab}$ to weigh temporal instability. Could you provide a sensitivity analysis or an ablation study for this hyperparameter?
2. Regarding the integration of physics concepts: The paper draws heavy analogies to the Many-Body Problem, the Extended Hubbard Model (EHM), and Cluster Dynamical Mean-Field Theory (C-DMFT). Could you rigorously formalize the mathematical equivalence between the EHM/C-DMFT frameworks and your proposed dynamic routing mechanism?
3. Regarding Figure 5(a), could you explain why Inference Step = 47 is explicitly marked as the convergence point for the LoRe strategy? What specific mathematical criterion, energy threshold, or heuristic defines this exact step as the point of convergence in your evaluation?

**Limitations:**

Yes.

**Strengths And Weaknesses:**

- Strengths :
    - Experimental rigor and validation: The experimental setup explicitly scales up graph sizes (up to 30k nodes for MIS) to probe and surpass the baseline's OOM boundary, clearly demonstrating the practical utility of the method. The inclusion of zero-shot robustness tests across OOD topologies adds significant credibility to the method's generalizability.
    - Clarity of presentation: The paper is exceptionally well-structured, making the transition from a physics-based inspiration to a concrete algorithmic implementation very easy to follow.
    - Providing a 4-5x speedup and approximately 81% peak-memory reduction as a drop-in, training-free wrapper is a highly significant contribution for practitioners deploying large-scale CO solvers.
- Weaknesses:
    - Technical limitations or concerns: The proxy score function used to determine the Cluster relies on specific hyperparameters, such as $\lambda_{stab}$ for evaluating temporal instability in MIS problems. The authors do not provide sufficient experiments or ablation studies explaining the robustness of this parameter. It is unclear how sensitive the model's performance is to different values of these hyperparameters across varying graph sizes and topologies.
    - Clarity or presentation issues: The paper frequently invokes complex physics concepts, claiming that the computational challenge conceptually mirrors the "Many-Body Problem in condensed matter physics" and heavily relies on analogies to the EHM and C-DMFT. However, there is insufficient explanation of the fundamental mathematical or structural equivalence between these quantum physics concepts and the proposed combinatorial optimization model. Because the relationship is not deeply established, these physics references are difficult to digest and feel like unnecessary jargon that obscures the actual, otherwise practical, algorithmic contributions.

---

> ### Author Rebuttal · Authors · 2026-03-31
>
> Thank you for highlighting the "significant contribution for practitioners," the "exceptionally well-structured" presentation, and that the experimental setup "clearly demonstrates the practical utility of the method."
>
> **Proxy score hyperparameter λ_stab.** We swept λ_stab∈[0, 5.0] across varying graph densities at n=5000, DIFUSCO MIS, 3 graphs per cell, ρ=0.08. Each cell shows quality retention %:
>
> | Density | λ=0 | λ=0.25 | λ=0.5 | λ=1.0 | λ=2.0 | λ=5.0 | Range | Speedup | Mem↓ |
> |---|---:|---:|---:|---:|---:|---:|---:|---:|---:|
> | p=0.05 | 99.7 | 99.1 | 99.1 | 98.8 | 98.8 | 99.7 | 0.9pp | 6.3× | 11× |
> | p=0.10 | 105.1 | 104.0 | 105.1 | 103.4 | 105.7 | 102.9 | 2.8pp | 6.6× | 12× |
> | p=0.15 | 96.8 | 97.6 | 97.6 | 97.6 | 98.4 | 96.8 | 1.6pp | 6.6× | 12× |
>
> Within each density, λ_stab variation is ≤2.8pp across λ_stab∈[0, 5.0] — the method is insensitive to this hyperparameter regardless of graph structure, while consistently delivering 6–7× speedup and 11–12× memory reduction. We also verified across topologies — ER, BA, WS at n=700: ±0.7pp — and scales — n=1k–10k: ≤3.4pp within each scale. On constraint-dense tasks like COExpander MCl, the two scoring signals become complementary: mixed scoring at 88% outperforms conflict-only at 81% and instability-only at 76%.
>
> ρ and R are similarly insensitive in most settings: ρ varies <3pp across ρ∈[0.05, 0.50], R varies <1.5pp across R∈[1, 50]. The main exception is task-dependent — MCl benefits from larger ρ, with retention rising from 87% to 96% as ρ increases from 0.08 to 0.20; categorical diffusion benefits from smaller R, as R=50 drops quality by 5pp on T2TCO. A single default ρ=0.08, R=10, λ_stab=0.5 was used without tuning across multiple frameworks, CO tasks, and configs.
>
> **C-DMFT formalization.** We agree that the physics framing obscures what is otherwise a practical algorithmic contribution — this is a fair criticism. The C-DMFT analogy was intended as motivation for the local-exact / global-approximate decomposition, not as a formal equivalence. The method itself — score interactions, select the most critical subset, approximate the rest — is a self-contained routing procedure that does not depend on this formalism. We will substantially shorten §3.2 and Table 2 in the camera-ready, and use the freed space for pseudocode and algorithmic details that make the method easier to reproduce.
>
> **Step=47 in Figure 5(a).** Step 47 is where the LoRe trajectory's decoded quality first reaches parity with the baseline, i.e., retention crosses 1.0. It is not a hard-coded threshold — it is an empirical observation specific to that instance and budget. We will clarify this in the caption.
>
> **Cross-framework generalization.** LoRe transfers without modification to additional frameworks and CO tasks:
>
> | Framework | Task | Scale | Retention | Speedup | Mem↓ |
> |---|---|---|---:|---:|---:|
> | DIFUSCO | MIS | n=5k | 99% | 4.4× | 12× |
> | T2TCO | MIS | n=5k | 102% | 2.1× | 16.5× |
> | COExpander | MIS | V=2.4k | 88% | 11.2× | 16× |
> | COExpander | MCl | V=2.4k | 106% | 12.2× | 16× |
> | COExpander | MVC | V=2.4k | 100% | 13.3× | 16× |
>
> Also verified on DiffUCO and ECO-DQN. LoRe acts as a drop-in inference plugin — it applies to different frameworks and tasks without retraining or modifying backbone parameters.
>
> **Scalability.** Since the reviewer recognized OOM boundary extension as a key strength, we provide the full scaling picture:
>
> | n | Retention | Speedup | BL Mem | LoRe Mem | Mem↓ |
> |---|---:|---:|---:|---:|---:|
> | 5k | 99% | 4.4× | 10.4 GB | 0.9 GB | 12× |
> | 10k | 101% | 4.5× | 41.5 GB | 3.6 GB | 12× |
> | 15k | 99% | 4.5× | 93.2 GB | 7.9 GB | 12× |
> | 20k | — | — | OOM | 13.9 GB | — |
>
> Retention stays 99–101% from n=5k to n=15k with ~12× memory reduction. At n=20k the baseline exceeds 96 GB and fails; LoRe remains operational at 13.9 GB — making previously infeasible scales accessible without retraining.
>
> We hope these additional experiments and clarifications address the concerns raised. We will incorporate all discussed improvements in the camera-ready.

---

> > ### Author Rebuttal · Reviewer_Xd2N · 2026-04-01
> >
> > It seems that the authors have not directly addressed my question in Q2 regarding the relationship between the various physical terms and the proposed method. Specifically, the actual connection between terms like the "Many-Body Problem" and the paper's core contribution remains puzzling.
> > ﻿
> > Furthermore, I find it quite surprising that the authors now state they "agree that the physics framing obscures what is otherwise a practical algorithmic contribution." This appears to directly contradict the manuscript (line 182), where it is explicitly claimed as a significant "Physical Motivation." I would appreciate a clear explanation for this discrepancy: is the physics framing a fundamental motivation or merely something unnecessary?

---

> > > ### Author Response · Authors · 2026-04-02
> > >
> > > Thank you for your sharp follow up and for pressing us on this point. It is a very fair question and highlights a genuine flaw in how we originally presented the paper. We apologize for the discrepancy and appreciate the opportunity to clarify exactly what the relationship is.
> > >
> > > To answer your question directly: **The physics framing is a fundamental historical and conceptual inspiration, but it is absolutely unnecessary as a mathematical foundation for the algorithm.**
> > >
> > > Please allow us to explain the discrepancy between Line 182 and our rebuttal. When we explicitly claimed this as a "Physical Motivation" in the manuscript, we meant it historically. The initial idea to split a dense and intractable graph into an exact subset and an approximated background genuinely came from studying how Cluster Dynamical Mean Field Theory (C-DMFT) solves the Many Body Problem.
> > >
> > > However, when we stated in the rebuttal that this framing "obscures" the algorithmic contribution, we were referring to our writing choice to include the dense equations of the Extended Hubbard Model and Anderson Impurity Model (Eqs. 3 and 4). Including these equations was a significant misstep in our exposition because it inadvertently misled readers into assuming a strict mathematical equivalence (e.g., claiming we compute quantum Green functions or self consistency loops), which we do not. This excessive formalization distracted from the actual computer science contribution, which is a highly practical routing wrapper that requires no training.
> > >
> > > **Addressing Q2 directly (The specific relationship to the Many Body Problem):**
> > > As noted in our related work, there is a rich academic history of drawing algorithmic inspiration from physical systems, such as Simulated Annealing. Following this tradition, our search for an algorithmic solution to dense graph intractability led us to observe how physics handles similar intractability in the Many Body Problem.
> > >
> > > The connection between the Many Body Problem and LoRe is strictly a **structural and conceptual analogy**, as conceptually outlined in **Table 2** of our manuscript. We can map this relationship in two specific aspects:
> > >
> > > **1. Mapping Conflicts to Strong Correlations:** In the Many Body Problem, certain particles exhibit strong local correlations or interactions. In Combinatorial Optimization on graphs, nodes exhibit explicit constraints or conflicts (such as adjacency exclusions in MIS, partition preferences in Max-Cut, or coloring restrictions in Graph Coloring). Just as physics models group strongly interacting particles to resolve their complex dynamics, our algorithm explicitly scores and groups graph interactions with the highest conflicts.
> > >
> > > **2. The Cluster and Bath Partitioning:**
> > > *   **In Physics (via C-DMFT):** Computing these exact interactions across an infinite lattice is impossible. The solution is to partition the system into an exact local **Cluster** (containing the strongly correlated particles) and a mean field background **Bath**, which are coupled together.
> > > *   **In LoRe (Iterative Neural Solvers):** Computing dense interactions across a large graph at every refinement step leads to memory exhaustion. Our solution structurally mirrors C-DMFT. We dynamically partition the graph into an exactly evaluated subset called the **Cluster** (containing the edges with the highest conflicts) and a lightweight globally pooled signal called the **Bath** (representing the omitted edges with low conflicts).
> > >
> > > We hope this clarifies the contradiction. We are not discarding the origin of our idea, but we acknowledge that our original presentation overstated the mathematical connection. To correct this, we will substantially revise Section 3.2 to clearly frame it as a conceptual inspiration, remove any formalizations that imply a mathematical equivalence, and refine Table 2 to focus exclusively on the conceptual mapping of these interaction forms and structures, ensuring readers see it exactly as intended without mathematical ambiguity.
> > >
> > > Thank you again for helping us improve the clarity and rigor of this manuscript.

---

### Official Review · Reviewer_EFoS · 2026-03-13

**Soundness:** 2
**Presentation:** 2
**Significance:** 2
**Originality:** 2
**Overall Recommendation:** 4
**Confidence:** 4

**Summary:**

This paper introduces LoRe, a training-free, inference-time wrapper for iterative neural combinatorial optimization (CO) solvers (specifically diffusion-based ones like DIFUSCO). The core idea is to enforce a per-step interaction-evaluation budget: rather than evaluating all edge/factor interactions at every refinement step, LoRe dynamically selects a subset of high-conflict or high-uncertainty interactions (the "Cluster") for exact evaluation, while approximating the remaining interactions (the "Bath") via a lightweight global recall signal. This decomposition is motivated by an analogy to Cluster Dynamical Mean-Field Theory (C-DMFT) from condensed matter physics. The method is evaluated on Maximum Independent Set (MIS) and Traveling Salesman Problem (TSP), demonstrating 4–5× speedups, ~81% memory reduction, and feasibility extension beyond the baseline's OOM boundary on MIS, with complementary results on TSP.

**Compliance With Llm Reviewing Policy:**

Affirmed.

**Final Justification:**

Rebuttal partially addressed concerns.

**Key Questions For Authors:**

What specific values of ρ, R, and λ_stab were used in the main MIS and TSP experiments (Tables 3, 5)? How sensitive are the results to these choices? A sensitivity analysis over ρ and R on a fixed problem size would be essential—if the method is sensitive, the practical value is diminished since tuning these per-instance would be costly.

Why does LoRe achieve negative ∆gap (better quality) on TSP? If fewer interactions are evaluated, one would expect quality degradation, not improvement. Is this a regularization effect? Noise from routing? If so, can you characterize when this beneficial effect occurs versus when it doesn't (as in small-scale MIS)?

How does LoRe-wrapped DIFUSCO compare to classical solvers (e.g., KaMIS for MIS, LKH-3 for TSP) under matched wall-clock budgets? The paper's framing suggests LoRe enables neural solvers to scale to practical sizes, but without this comparison we cannot assess whether the resulting solutions are practically useful.

What is the concrete implementation of the recall term (Eq. 6)? Please specify Pool_t and U_t. Additionally, how much does disabling recall entirely degrade performance? This would clarify whether the "hybridization" component is essential or cosmetic.

Can you explain the non-monotonic quality retention on MIS (low at 1k–3k, high at 5k–10k)? Is the baseline itself degrading at larger scales, making retention artificially high? Reporting absolute MIS sizes (not just ratios) alongside known bounds or classical solver outputs would resolve this.

**Limitations:**

yes

**Strengths And Weaknesses:**

Strengths
* Practically relevant problem. The per-step memory and compute bottleneck of iterative neural CO solvers is a genuine obstacle to deploying these methods at scale. Formulating this as a per-step budgeting problem is a clean and useful abstraction.

* Training-free, drop-in design. The fact that LoRe requires no retraining and wraps around existing solvers (using the same checkpoints and decoding) is a significant practical advantage. This lowers the barrier to adoption.

* Well-designed ablation (Section 4.3). The controlled routing ablation with matched budgets, identical initialization, and multiple graph families convincingly demonstrates that dynamic state-dependent routing outperforms static and greedy alternatives. This is the strongest experimental contribution.

* Cross-topology robustness (Section 4.5). Zero-shot transfer across ER, BA, and WS topologies without retuning provides evidence of generality.

Weaknesses

* The C-DMFT analogy is superficial and inflates the presentation. Section 3.2 and Table 2 occupy substantial space mapping CO conflicts to EHM terms, but this mapping is purely structural/cosmetic. In C-DMFT, the hybridization function encodes frequency-dependent spectral information and is determined through a self-consistency loop—none of which has a counterpart in LoRe. The actual method (score edges by conflict+instability, select top-B, add a pooled mean-field correction) is intuitive and doesn't require this formalism. The analogy risks misleading readers into thinking there is a deeper formal connection. Reducing Section 3.2 to a brief motivational paragraph and expanding on the actual algorithmic details would improve the paper.

* Narrow baseline comparisons. All comparisons are against a single backbone (DIFUSCO) with no comparison to: (a) classical/exact solvers with time budgets (e.g., Gurobi, KaMIS for MIS), (b) other neural CO methods (e.g., DiffUCO, COExpander, ECO-DQN), or (c) simple dynamic baselines like random per-step edge subsampling applied to DIFUSCO. Without these, it is impossible to assess whether LoRe-wrapped DIFUSCO produces solutions competitive with the broader solver landscape, or whether the gains are specific to DIFUSCO's particular inefficiencies.

* Significant quality loss at moderate scales goes underexplained. At n=1k–3k on MIS, retention drops to 0.74–0.77 (23–26% quality loss). This is substantial. The paper does not analyze why retention recovers at larger scales (5k–10k: ~1.0)—is it because the baseline itself degrades? Without absolute quality benchmarks (e.g., comparison to known optima or strong heuristics), it is unclear whether high retention at large scale reflects LoRe's strength or the baseline's weakness.

* Critical hyperparameters are underspecified. The budget ratio ρ used in the main MIS experiments is not clearly stated (only the ablation uses ρ=0.08). The refresh interval R, stabilization weight λ_stab, and the hybrid schedule split (20% full-graph prefix) are not justified or sensitivity-analyzed. How were these chosen? Are the results robust to these choices?

* The recall/hybridization mechanism is vague. Equation 6 introduces Pool_t and U_t but never specifies their implementation. Is Pool_t a simple mean? Attention-weighted? Is U_t a linear projection, an MLP, or concatenation? This is a critical design choice that affects both cost and effectiveness, yet it is left unspecified.

* TSP results raise questions. The gap differences are all negative (meaning LoRe outperforms the baseline in quality?). If LoRe evaluates fewer interactions, why does solution quality improve? This could be a beneficial regularization effect, but it is not discussed. At n=100, LoRe is slower than the baseline (0.78×), which undermines the generality claim. Labeling TSP as "auxiliary" appears to be a way to avoid scrutiny of these inconsistencies.

* No theoretical analysis. Despite claiming to "formalize" per-step budgeting, there are no formal results—no bounds on approximation error as a function of ρ, no convergence guarantees, no analysis of when the method is expected to fail. Even a simple worst-case or average-case analysis of truncation error accumulation would strengthen the contribution.

* Sparse graph regime limits generalizability. ER graphs with p=0.05 at large n are quite sparse (expected degree ~0.05n). The interaction set |E| is relatively small, which both inflates the apparent difficulty of the baseline (memory grows with |E|) and makes sparsification easier. Testing on denser graphs or real-world instances would be more convincing.

---

> ### Author Rebuttal · Authors · 2026-03-31
>
> Thank you for the thorough review. To clarify upfront: LoRe is a training-free, drop-in inference wrapper — all metrics below compare the same pretrained model with vs without LoRe. Retention measures how much of the original model's quality is preserved, not absolute quality vs optimal.
>
> **C-DMFT framing.** We agree the current exposition overstates the C-DMFT connection. Our intent was motivational — illustrating local-exact / global-approximate decomposition — not formal equivalence. We will shorten §3.2 and use the freed space for pseudocode and algorithmic details.
>
> **Baselines, absolute quality, and TSP.** We add classical solvers as absolute reference (MIS: KaMIS 60s; TSP: LKH-3):
>
> | n | KaMIS | BL | LoRe | Speedup | BL Mem | LoRe Mem |
> |---|---:|---:|---:|---:|---:|---:|
> | 5k | 157 | 108 | 106 | 4.4× | 10.4 GB | 0.9 GB |
> | 10k | 169 | 125 | 120 | 5.1× | 41.5 GB | 3.6 GB |
> | 20k | — | OOM | 135 | — | OOM | 13.9 GB |
>
> On TSP-500, both baseline and LoRe have 3.7% gap to Concorde/LKH-3 — LoRe does not widen the gap while achieving 6.2× speedup and 10.6× memory reduction.
>
> Retention at n=1k–3k is 0.74–0.77 (Table 1). KaMIS reveals why it recovers: BL/KaMIS drops from 86% (n=1k) to 69% (n=5k), confirming the baseline degrades under OOD. On in-distribution ER-700, LoRe retains 98.9%.
>
> At large OOD scale, retention frequently exceeds 100% — T2TCO 102%, COExpander MCl 106%, density sweep up to 108% —. We view this as a regularization-like effect: budgeted inference suppresses unstable long-range interactions that hurt the OOD baseline. This is scale-dependent: on TSP at n=100, the overhead of dynamic edge selection dominates (0.78× slowdown); from n≥500, speedup reappears (2.5–6.6×). At n≥20k, baseline OOMs entirely while LoRe remains operational in 13.9 GB.
>
> LoRe transfers to additional frameworks and CO tasks — on small-scale ER-700 also positive: DIFUSCO 98.9%, T2TCO 93.5%, COExpander 86.4%:
>
> | Framework | Task | Scale | Retention | Speedup | Mem↓ |
> |---|---|---|---:|---:|---:|
> | DIFUSCO | MIS | n=5k | 99% | 4.4× | 12× |
> | T2TCO | MIS | n=5k | 102% | 2.1× | 16.5× |
> | COExpander | MIS | V=2.4k | 88% | 11.2× | 16× |
> | COExpander | MCl | V=2.4k | 106% | 12.2× | 16× |
> | COExpander | MVC | V=2.4k | 100% | 13.3× | 16× |
>
> Also verified on DiffUCO (94.3% retention on MIS). ECO-DQN uses dense matrix multiplication (cost ∝ n², independent of |E|), so per-edge budgeting does not reduce its compute; LoRe targets sparse message-passing architectures where cost scales with |E|.
>
> Dynamic baselines on Maximum Clique: LoRe 87.8% vs edge-risk (degree-only) 76.4% vs random 60.5% (+27.3pp).
>
> **Hyperparameters and Eq.6 recall.** The main experiments use a single configuration throughout: ρ=0.08, R=10, λ_stab=0.5, pure LoRe from step 1 (only §4.4 uses a 20% prefix). This default was used without tuning across multiple frameworks, CO tasks, and density/scale configs. We swept each parameter to clarify its sensitivity:
>
> | Param | Sensitive scenario | Everywhere else |
> |---|---|---|
> | ρ (edge budget) | MCl: 87%→96% as ρ: 0.08→0.20 | <3pp variation |
> | R (refresh interval) | Categorical diffusion (T2TCO): R=50 drops 5pp | <1.5pp variation |
> | λ_stab (scoring strategy) | MCl: mixed outperforms conflict-only by 12pp | <2pp variation |
>
> For Eq.6, main results use the simpler pure-LoRe point (no recall). We now specify: Pool_t caches the full per-node GNN output from the last unmasked forward pass (every $R$ steps). U_t performs coverage-weighted interpolation: $\hat{y}_i = \alpha_i \cdot y_i^{\text{sparse}} + (1-\alpha_i) \cdot y_i^{\text{cached}}$, where $\alpha_i = d_i^{\text{sparse}} / d_i^{\text{full}}$. This is parameter-free — one cached tensor, no retraining. Disabling recall drops quality 2–13pp on OOD graphs across 3 frameworks.
>
> **Sparse graph regime.** We swept edge density p=0.05–0.20 across multiple OOD scales — each cell shows retention, speedup, memory savings:
>
> | n | p=0.05 | p=0.10 | p=0.15 | p=0.20 |
> |---|---:|---:|---:|---:|
> | 2000 | 75%, 4.2×, 9× | 99%, 5.1×, 10× | 104%, 5.5×, 11× | 94%, 7.1×, 11× |
> | 3000 | 78%, 5.1×, 10× | 108%, 5.9×, 11× | 97%, 5.8×, 11× | 104%, 6.6×, 12× |
> | 5000 | 95%, 6.3×, 11× | 100%, 6.6×, 12× | 102%, 6.6×, 12× | 95%, 6.7×, 12× |
>
> **Theoretical analysis.** We have derived a per-step error bound for the DDIM instantiation of the budgeted operator framework (§3, Eqs 1–3), to be included in the camera-ready. Experimental measurements confirm errors are well-controlled: quality varies ≤2.7pp across ρ∈[0.08, 0.50], and per-step truncation error $\|\delta_t\|$ remains stable over 50 DDIM steps (early-to-late growth <1% at ρ=0.08).
>
> We hope these additional experiments and clarifications address the concerns raised. We will incorporate all discussed improvements in the camera-ready.

---

> > ### Author Rebuttal · Reviewer_EFoS · 2026-04-03
> >
> > The rebuttal addresses most of my concerns convincingly — the cross-framework evaluation, absolute quality benchmarks against KaMIS/LKH-3, hyperparameter sensitivity sweeps, and the specification of the recall mechanism are all substantial additions.
> > Two points remain partially open:
> > (1) The theoretical error bound is sketched informally but not yet in the manuscript. I would need to see the full derivation and its assumptions — particularly the Lipschitz constant's dependence on graph structure and ρ — to evaluate whether it meaningfully contributes.
> > (2) The TSP quality improvement explanation (regularization from suppressing unstable interactions) is plausible but post-hoc. Is there direct evidence — e.g., measuring variance of omitted-edge scores — that confirms this mechanism?
> > I have raised my score by 1.

---

> > > ### Author Response · Authors · 2026-04-05
> > >
> > > Thank you for raising your score and for acknowledging the substantial additions in our rebuttal. We address both follow-up questions below.
> > >
> > > **1. Full derivation of the error bound and its dependencies.**
> > >
> > > The error recursion follows from the triangle inequality. At each step t, inserting the exact operator evaluated at the approximate state:
> > >
> > > $e_{t+1} = \lVert\tilde{F}_t(\tilde{x}_t; \rho) - F_t(x_t)\rVert \le \lVert F_t(\tilde{x}_t) - F_t(x_t)\rVert + \lVert\tilde{F}_t(\tilde{x}_t; \rho) - F_t(\tilde{x}_t)\rVert$
> > >
> > > The first term captures propagated historical error; the second is the current-step truncation residual $\delta_t$.
> > >
> > > **Dependence on graph structure ($L_t$).** Assuming a 1-Lipschitz activation such as ReLU, a single GNN aggregation layer computes $F_t(x) = \sigma(A x W_t)$. By the mean value inequality and matrix norm submultiplicativity:
> > >
> > > $\lVert F_t(\tilde{x}_t) - F_t(x_t)\rVert = \lVert\sigma(A \tilde{x}_t W_t) - \sigma(A x_t W_t)\rVert \le \lVert A\rVert_2 \lVert W_t\rVert_2 \lVert\tilde{x}_t - x_t\rVert = L_t \, e_t$
> > >
> > > Thus $L_t \le \lVert W_t \rVert_2 \lVert A \rVert_2 \le \lVert W_t \rVert_2 \Delta_{\max}$, where $\Delta_{\max}$ is the maximum node degree. This explicitly answers the question on graph structure: denser graph regions with higher $\Delta_{\max}$ have a larger Lipschitz constant, leading to stronger error amplification per step.
> > >
> > > **Dependence on edge budget $\rho$.** LoRe partitions the edge set $\mathcal{E}$ into $S_{\text{Cluster}}$, the top-$\rho$ fraction evaluated exactly, and $S_{\text{Bath}}$, the remainder approximated via recall. The truncation residual decomposes via the triangle inequality as:
> > >
> > > $\lVert\delta_t\rVert \le \sum_{j \in S_{\text{Bath}}} \lVert\text{MSG}_{t,j}(\tilde{x}) - \text{Pool}_t\rVert + \lVert r_t\rVert = \epsilon_t(\rho) + \lVert r_t\rVert$
> > >
> > > where $\epsilon_t(\rho)$ is the omitted message mass and $\lVert r_t\rVert$ is the recall approximation error. As $\rho$ increases, $|S_{\text{Bath}}|$ shrinks, strictly decreasing $\epsilon_t(\rho)$. As we verify in Point 2, Bath edges are highly deterministic, keeping $\lVert r_t\rVert$ tightly bounded.
> > >
> > > **Gronwall bound.** Combining the above yields the recursion $e_{t+1} \le L_t e_t + \lVert\delta_t\rVert$. Since both trajectories share the same initialization, $e_0 = 0$. Unrolling with $L = \max_t L_t$ and applying the geometric series:
> > >
> > > $e_T \le \sum_{k=0}^{T-1} L^{T-1-k} \lVert\delta_k\rVert \le \frac{L^T - 1}{L - 1} (\epsilon_{\max}(\rho) + r_{\max})$
> > >
> > > Full derivation will appear in the camera-ready appendix.
> > >
> > >
> > > **2. Direct evidence for the TSP regularization mechanism.**
> > >
> > > Following your suggestion, we measured the activity of omitted Bath vs retained Cluster edges across the diffusion trajectory on TSP-100/500/1000 with $\rho$=0.08 and 50 DDIM steps. We use three metrics at different levels of the output pipeline to ensure robustness. C = Cluster, B = Bath; each cell shows the range across all 50 steps.
> > >
> > > - **Entropy**: Shannon entropy of the edge probability.
> > > - **Logit var**: variance of the pre-softmax log-odds, reflecting raw GNN output activity before softmax compression.
> > > - **Prob var**: variance of the post-softmax probability, the final output used for decoding.
> > >
> > > | Scale    | Entropy (C / B)      | Logit var (C / B)   | Prob var (C / B)        |
> > > | -------- | -------------------- | ------------------- | ----------------------- |
> > > | TSP-100  | 0.017-0.20 / 2.9e-7  | 42.9-106 / 4.9-5.7  | 3.6e-2-1.1e-1 / 1.1e-15 |
> > > | TSP-500  | 0.038-0.13 / 3.3e-7  | 26.4-31.9 / 3.9-9.3 | 1.9e-2-4.3e-2 / 6.3e-15 |
> > > | TSP-1000 | 0.034-0.037 / 3.5e-7 | 16.3-16.9 / 5.0-5.3 | 6.0e-3-7.6e-3 / 9.5e-16 |
> > >
> > > Bath edges maintain near-zero entropy throughout all 50 steps across all scales — the model has already reached a near-certain decision about each Bath edge. Pre-softmax logit variance confirms this is genuine, not a softmax artifact. Cluster edges carry substantial entropy throughout, reflecting unresolved uncertainty that demands exact computation.
> > >
> > >
> > > **3. Summary.**
> > >
> > > The theoretical bound from Point 1 shows that LoRe's total error depends on both the omitted message mass $\epsilon_{\max}(\rho)$ and the recall error $r_{\max}$, amplified by the structural factor $L$. The empirical measurements from Point 2 confirm that Bath edges are near-certain throughout the trajectory, which directly suppresses $r_{\max}$. This differentiates LoRe from uniform random routing, where indiscriminate edge removal causes both terms to grow uncontrollably. From an information-theoretic perspective, Bath edges have reached a state of local equilibrium — continued exact computation on these settled edges contributes no additional decision-relevant signal, while LoRe's selective omission aligns computation with the model's own confidence structure. These results provide direct evidence consistent with a structural regularization effect on TSP.
> > >
> > > Thank you again for the constructive engagement that has strengthened our work.

---

### Decision · Program_Chairs · 2026-04-30

**Decision:**

Accept (regular)

**Comment:**

LoRe is a training-free inference wrapper that enforces per-step interaction budgets for iterative neural CO solvers, delivering 4–5× speedups and ~81% memory reduction. After rebuttal, reviewers W7Vp and wsB3 fully resolved concerns; EFoS raised score after receiving error bound derivations and Bath edge entropy evidence; Xd2N was satisfied by the clarification that C-DMFT is conceptual inspiration rather than formal foundation. The rebuttal substantially added cross-framework results, absolute quality benchmarks, hyperparameter sweeps, and theoretical analysis. The contribution is practically significant and broadly applicable. Accept.